# Formation of individual stripes in a mixed-dimensional cold-atom Fermi–Hubbard system

Dominik Bourgund[1,2 ✉], Thomas Chalopin[1,2], Petar Bojović[1,2], Henning Schlömer[2,3,4], Si Wang[1,2], Titus Franz[1,2], Sarah Hirthe[1,2,6], Annabelle Bohrdt[2,5], Fabian Grusdt[2,3,4], Immanuel Bloch[1,2,4] & Timon A. Hilker[1,2 ✉]

The relation between $d$-wave superconductivity and stripes is fundamental to the understanding of ordered phases in high-temperature cuprate superconductors[1–6]. These phases can be strongly influenced by anisotropic couplings, leading to higher critical temperatures, as emphasized by the recent discovery of superconductivity in nickelates[7–10]. Quantum simulators with ultracold atoms provide a versatile platform to engineer such couplings and to observe emergent structures in real space with single-particle resolution. Here we show, to our knowledge, the first signatures of individual stripes in a cold-atom Fermi–Hubbard quantum simulator using mixed-dimensional (mixD) settings. Increasing the energy scale of hole–hole attraction to the spin exchange energy, we access the interesting crossover temperature regime in which stripes begin to form[11]. We observe extended, attractive correlations between hole dopants and find an increased probability of forming larger structures akin to individual stripes. In the spin sector, we study correlation functions up to the third order and find results consistent with stripe formation. These observations are interpreted as a precursor to the stripe phase, which is characterized by interleaved charge and spin density wave ordering with fluctuating lines of dopants separating domains of opposite antiferromagnetic order[12–14].

The phase diagram of high-temperature superconducting materials has so far eluded full understanding despite 40 years of extensive theoretical and experimental studies[4–6]. Especially, the exact nature of the intricate relationship between superconducting pairs and stripes remains an open question[1–6]. Although experimentally both phenomena may be found in close proximity, numerical studies have long been investigating whether stripes precede, compete or coexist with superconductivity[15–17]. Recently, by the discovery of superconductivity in bilayer nickelates[7], a new class of superconducting materials has been found. These materials are conjectured to be of a mixD nature, in which the dynamics of charge is restricted to two dimensions, whereas the spins order in a bilayer system. Here we study mixD systems, which are predicted to show an enhanced version of stripe order[8–10].

The repulsive, two-dimensional (2d), spin-1/2 Fermi–Hubbard model and its natural extensions are widely assumed to provide minimal models to capture the physics of these doped antiferromagnets (AFMs). Ultracold atoms in optical lattices provide natural implementations of the Hubbard model with a high degree of control over system parameters and dimensionality[18,19]. Although solid-state experiments mostly focus on spectroscopic and dynamical response measurements,

quantum simulation, especially with single-site resolution, opened up access to new sets of microscopic observables and correlation functions[20–23]. Previous studies in 2d systems found AFM correlations[24–29], investigated the effect of doping on the spin order[19,30–35] and observed pairing of dopants in tailored ladder systems[36].

Here we present the first, to our knowledge, observation of hole attraction beyond nearest-neighbouring sites in a repulsive, 2d Hubbard system with mixD coupling. Using higher-order charge and spin correlators, we find signatures of extended charge structures, which we identify as individual stripes.

## Stripe formation and mixD

Stripe phases, characterized by charge density waves in combination with incommensurate AFM order, have been found in measurements on solids[1–3,37,38] as well as numerical studies[12–14,17,39–43]. These stripes form out of individual dopants of an AFM background, a process governed by the competition between the kinetic energy favouring delocalization and the magnetic energy of the AFM spin order, which is disrupted by dopant motion. Consequently, the energy scale at which stripe order is expected to occur is only around 5% of the tunnelling energy[43]

[1]Max-Planck-Institut für Quantenoptik, Garching, Germany. [2]Munich Center for Quantum Science and Technology, Munich, Germany. [3]Arnold Sommerfeld Center for Theoretical Physics (ASC), Ludwig-Maximilians-Universität, Munich, Germany. [4]Fakultät für Physik, Ludwig-Maximilians-Universität, Munich, Germany. [5]Institute of Theoretical Physics, University of Regensburg, Regensburg, Germany. [6]Present address: ICFO - Institut de Ciencies Fotoniques, The Barcelona Institute of Science and Technology, Castelldefels, Spain. ✉e-mail: dominik.bourgund@mpq.mpg.de; timon.hilker@mpq.mpg.de

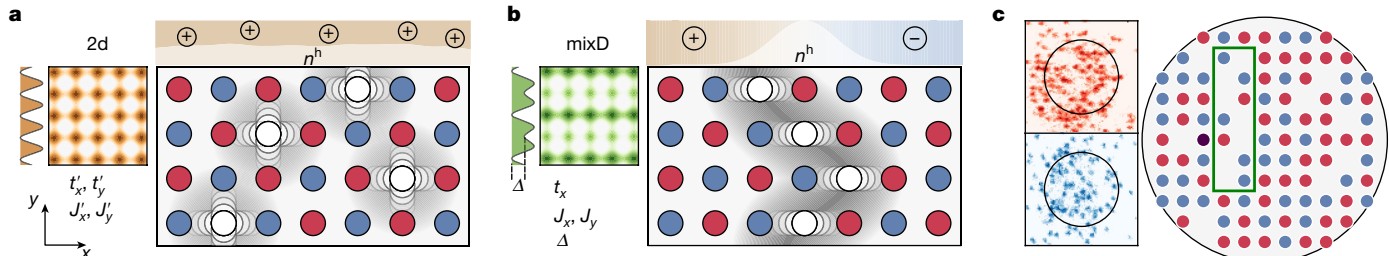

**Fig. 1 | MixD Fermi–Hubbard systems. a**, Illustration of the isotropic 2d Fermi–Hubbard model. Holes delocalize within small regions and disturb their respective spin background, forming magnetic polarons. The overall hole density is uniform and holes repel each other owing to their fermionic statistics at experimentally accessible temperatures of $k_B T \approx J$. There are no domain walls in the spin order. **b**, By raising the potential on every other lattice site along $y$ by $\Delta$, we suppress tunnelling along this direction, thus removing the Pauli repulsion between holes, while preserving the superexchange coupling $J_y$.

At low temperatures, the holes form collective structures, which also result in a domain wall in the AFM correlations of the system, indicated by the AFM parity change across the stripe. **c**, A single raw experimental shot of spin-up (red) and spin-down (blue) atoms and doubly occupied sites (purple), as well as its reconstructed spin and charge distribution with the main system being inside the black circle, surrounded by a low-density reservoir (see Supplementary Information). The green box indicates a stripe-like structure.

(about 10% of the superexchange energy), placing it out of reach for state-of-the-art quantum simulators. In particular, for temperatures around the superexchange energy, the effective repulsion resulting from the fermionic nature of the holes (Pauli blocking) disfavours tightly bound hole pairs and extended structures such as stripes but favours the formation of magnetic polarons[30] (see Fig. 1a).

MixD systems allow to tilt the balance towards collective charge and spin ordering by restricting the hole motion to one dimension, thus reducing the kinetic energy while keeping spin couplings 2d. This leads to an increase in the characteristic energy scales of collective effects as kinetic and magnetic terms in the Hamiltonian are less frustrated, lifting these effects to experimentally accessible regimes[44]. In nickelates, which are mixD bilayer systems, this causes high critical temperatures for superconductivity[7–10]. Similarly, in ladder systems, in which the spin order is dominated by rung singlets, numerics[11] and experiments[36] confirmed strong pairing in mixed dimensions. Here we apply this mixD approach to the unexplored 2d system, in which the spin sector is not gapped and classical simulations are limited to very small system sizes, especially at the relevant intermediate temperature scales. By adding a sufficiently large potential offset to every other chain within the lattice, we remove nearest-neighbour hopping along the perpendicular direction while increasing spin couplings[45,46]. This biases hole attraction and stripe formation along the direction perpendicular to the hole motion because Pauli repulsion is suppressed in this direction. For the same reason, fully filled stripes are favoured but the key concept of charge ordering associated with the stripe phase is retained[42,43] (see Fig. 1b). This allows us to study the poorly understood temperature regime around the superexchange energy in which individual stripes are expected to form.

## Experimental implementation

In the experiment, we realize the spin-1/2 Fermi–Hubbard model by using $^6$Li atoms in an optical superlattice with a homogeneous, circular system of about 110 sites surrounded by a low-density reservoir (see Fig. 1c). In the limit of strong on-site interactions $U$, the essential physics of the system can be captured by the $t$–$J$ Hamiltonian using projections $\hat{\mathcal{P}}$ onto singly occupied sites,

$$\hat{\mathcal{H}}_{t-J} = \sum_{\langle \mathbf{i},\mathbf{j}\rangle,\sigma} \hat{\mathcal{P}}(-t_{\mathbf{ij}}\hat{c}_{\mathbf{i},\sigma}^\dagger \hat{c}_{\mathbf{j},\sigma} + \text{h.c.})\hat{\mathcal{P}} + \sum_{\langle \mathbf{i},\mathbf{j}\rangle} J_{\mathbf{ij}}\left(\hat{\mathbf{S}}_{\mathbf{i}} \cdot \hat{\mathbf{S}}_{\mathbf{j}} - \frac{\hat{n}_{\mathbf{i}}\hat{n}_{\mathbf{j}}}{4}\right), \quad (1)$$

with tunnel couplings $t_{\mathbf{ij}} \in \{t_x, t_y\}$, spin exchange couplings $J_{\mathbf{ij}} \in \{J_x, J_y\}$, spin-up/spin-down fermionic creation (annihilation) operators on site $\mathbf{i}$, $\hat{c}_{\mathbf{i},\uparrow/\downarrow}^\dagger$ ($\hat{c}_{\mathbf{i},\uparrow/\downarrow}$) and on-site spin (density) operators $\hat{\mathbf{S}}_{\mathbf{i}}$ ($\hat{n}_{\mathbf{i}}$). This model

suffers from the fermion sign problem, making it numerically challenging to tackle even in the mixD regime[47].

Here we work at $U/t_x = 27(2)$, $J_y/t_x = 0.6(2)$, $J_x/t_x = 0.15(3)$ and a filling of $n \approx 0.7$–$0.9$ (hole doping $\delta = 1 - n$) with a temperature of $k_B T/t_x = 0.3(1)$ (see Supplementary Information). We make use of an optical superlattice along $y$ to controllably detune neighbouring sites by $\Delta = 0.65(5)U \gg t_x, t_y'$, thus effectively disabling nearest-neighbour tunnelling along $y$ ($t_y \approx 0$), and leading to a spin coupling $J_y = 2(t_y')^2\left(\frac{1}{U-\Delta} + \frac{1}{U+\Delta}\right)$, in which $t_y'$ is the tunnel coupling in the 2d system without potential offsets. Owing to the staggered superlattice potential, there is also a second-order next-nearest-neighbour hopping term along $y$, which reintroduces a weak Pauli repulsion at distance $d_y = 2$. This term, however, is smaller than $J_y$, such that it is still expected to be favourable for stripes to form (see Supplementary Information for more details on preparation and subdominant couplings).

## Hole–hole correlations

To reveal the charge order within the system, we evaluate the connected, normalized two-point hole–hole correlator

$$g_{hh}^{(2)}(\mathbf{d}) - 1 = \frac{1}{\mathcal{N}_\mathbf{d}} \sum_{\mathbf{i}} \left(\frac{\langle \hat{n}_\mathbf{i}^h \hat{n}_{\mathbf{i+d}}^h\rangle}{\langle \hat{n}_\mathbf{i}^h\rangle\langle \hat{n}_{\mathbf{i+d}}^h\rangle} - 1 - o_\delta\right), \quad (2)$$

with hole density operator $\hat{n}_\mathbf{i}^h$ at position $\mathbf{i}$ and normalization $\mathcal{N}_\mathbf{d}$ the number of bonds with distance $\mathbf{d}$. Owing to the finite size and particle number fluctuations in our system, there is a global, doping-dependent offset $o_\delta \in [-0.06, -0.03]$ on this correlator that we subtract (see Supplementary Information). A positive (negative) value of this correlator indicates attraction (repulsion) between holes at distance $\mathbf{d}$.

We consider hole correlations in a mixD system with a doping of $\delta = 0.18$ in Fig. 2a,b. We observe a positive nearest-neighbour correlation along $y$, whereas along $x$, we find antibunching caused by the Pauli repulsion of the holes (see Fig. 2a). Furthermore, at larger distances $d_y > 1$, there are positive correlations, which indicates that, instead of merely forming isolated, nearest-neighbour hole pairs, there is a finite probability that vertically aligned hole structures are extended through the system. Furthermore, there are significant correlations along the diagonals at $\mathbf{d} = (1, 1)$, which we interpret as signs of charge fluctuations along $x$ owing to the finite coupling $t_x$. The correlations at $d_y = 2$ are slightly suppressed, which we attribute to next-nearest-neighbour hopping (see Supplementary Information). Finally, the positive signal at $d_x = \pm 5$ may be related to the presence of a second, vertically aligned charge structure in the system.

By considering one-dimensional (1d) cuts along $y$ and $x$ (Fig. 2b), we corroborate the bunching (antibunching) along $y$ ($x$) through the

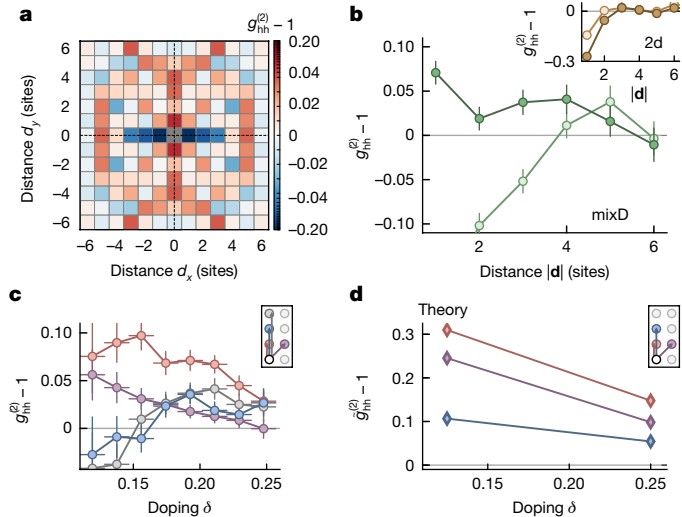

**Fig. 2 | Hole correlations beyond nearest neighbours. a**, Hole–hole correlations in a mixD system revealing bunching (attraction) along $y$ at distances $d_y \geq 1$ and antibunching (repulsion) along $x$ with $\delta = 0.18$. The symmetrization is indicated by the dashed lines. **b**, A cut along $y$ ($x$) is shown in dark (light) green for mixD systems, with the inset showing the equivalent data for standard 2d systems. **c**, The dependency of the mixD correlator $g_{hh}^{(2)}$ at distance **d** = (0, 1), (1, 1), (0, 2), (0, 3) on doping is plotted in red, purple, blue and grey with a doping binning of ±0.009. In a doping region around 0.2, the correlators for distances $d_y > 1$ are positive, indicating longer range charge correlations. Error bars are estimated using bootstrapping. **d**, Results for the renormalized correlator (see Supplementary Information) from DMRG calculations for a system size $L_x \times L_y = 8 \times 3$ as a function of doping for $k_B T/t_x = 0.41$.

system in the mixD setting. By contrast, for the standard 2d system ($\Delta = 0$; Fig. 2b, inset), there is antibunching along both directions. Compared with this, the anticorrelations along $x$ are enhanced in mixD by removing $t_y$ as a result of the absent competition between anticorrelations along $x$ and $y$.

To identify whether an ideal doping level for the emergence of individual stripes exists in our mixD system, we bin our data by doping and calculate $g_{hh}^{(2)}$ per bin (see Fig. 2c). Both the nearest-neighbour and diagonal correlations decrease with doping above $\delta \approx 0.15$, indicative of the decrease in pairing probability with doping and compatible with a reduction of the spin correlations responsible for the binding. For **d** = (0, 2), (0, 3), there is a non-trivial dependence on doping with positive correlator values starting at $\delta = 0.17$. This is indicative of a possible transition from the formation of individual pairs to extended stripe-like structures[41–43].

We compare the correlations along $y$ to density matrix renormalization group (DMRG) calculations of equation (1) on $8 \times 3$ sites, $J_y/t_x = 0.5$, $J_x/t_x = 0.15$, $k_B T/t_x = 0.41$ as a function of doping in Fig. 2d (see also Supplementary Information for normalization). Although the trend in correlations at **d** = (0, 1) and **d** = (1, 1) is qualitatively comparable with the experimental data, the same cannot be stated about correlations at **d** = (0, 2). We attribute these differences to the strong finite size limitations in the DMRG along $y$ as well as the simulation of the $t$–$J$ model instead of the Hubbard model. Further differences could arise owing to the presence of the aforementioned second-order hopping process that introduces further repulsion between holes, as well as the statistical distribution of holes between different chains in the experiment, whereas calculations feature balanced hole numbers.

## Structures beyond hole pairing

The connected two-point correlator $g_{hh}^{(2)}$ only provides limited insights into the physics of extended charge structures and how they interact

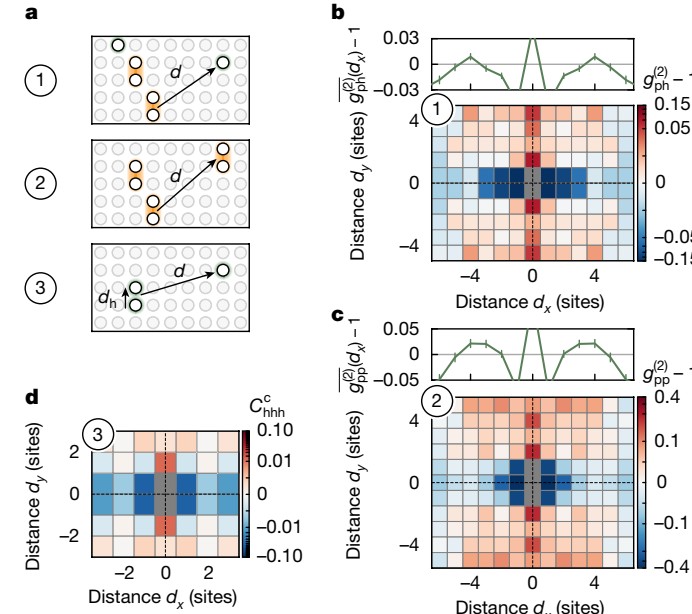

**Fig. 3 | Multipoint correlators. a**, Illustration of pair–hole, pair–pair and hole–hole–hole correlator, in which a pair is defined as a nearest-neighbour pair of holes along $y$. **b,c**, Symmetrized correlation map of the pair–hole (pair–pair) correlator. We find an attraction of the pairs along $y$, which points towards the formation of larger-scale structures. Above the map, the average over $d_y$ ($\overline{g^{(2)}}$) hints at the existence of another charge structure at $d_x = 4$. Error bars are estimated using bootstrapping and are smaller than the marker size if not visible. **d**, In the symmetrized, connected three-point hole–hole–hole correlator with $\mathbf{d}^h = (0, 1)$, we observe a positive signal at nearest neighbours along $y$, which indicates the existence of longer charge structures beyond pairs of two holes (see Supplementary Information for statistical significance). The data are evaluated over the hole doping distribution as given in the Supplementary Information.

with each other. We extend the analysis by considering the two-point pair–hole and pair–pair correlators

$$
\begin{cases}
g_{ph}^{(2)}(\mathbf{d}) - 1 = \dfrac{1}{\mathcal{N}_\mathbf{d}} \sum_\mathbf{i} \left( \dfrac{\langle \hat{n}_\mathbf{i}^p \hat{n}_{\mathbf{i+d}}^h \rangle}{\langle \hat{n}_\mathbf{i}^p \rangle \langle \hat{n}_{\mathbf{i+d}}^h \rangle} - 1 \right), \\
g_{pp}^{(2)}(\mathbf{d}) - 1 = \dfrac{1}{\mathcal{N}_\mathbf{d}} \sum_\mathbf{i} \left( \dfrac{\langle \hat{n}_\mathbf{i}^p \hat{n}_{\mathbf{i+d}}^p \rangle}{\langle \hat{n}_\mathbf{i}^p \rangle \langle \hat{n}_{\mathbf{i+d}}^p \rangle} - 1 \right),
\end{cases} \quad (3)
$$

in which we define the pair operator $\hat{n}_\mathbf{i}^p = \hat{n}_{(i_x,i_y)}^h \hat{n}_{(i_x,i_y+1)}^h$. These correlators, as shown in Fig. 3a, assume the existence of nearest-neighbour pairs along $y$ (established using $g_{hh}^{(2)}$) and consider the attraction or repulsion of these pairs to other dopants or pairs. They may be seen as fully connected and normalized two-point correlators of pairs or 'partially connected' three-point/four-point hole correlators. Note that, for simplicity, we neglect diagonal pairs (that is, $\hat{n}_{(i_x,i_y)}^h \hat{n}_{(i_x+1,i_y+1)}^h$) associated with fluctuations along $x$ and may thus underestimate the amount of order within the system.

We present the pair–hole and pair–pair correlations for the mixD system as a function of distance along $x$ and $y$ in Fig. 3b,c. For improved statistics, we include in our analysis all hole doping levels (see Supplementary Information) for which the offset $o_\delta$ of equation (2) becomes negligible. In both cases, we observe positive correlations along $y$, which extend throughout the system, indicating that individual pairs are not repelled from other holes or each other but instead align along $y$ and tend to form stripe-like structures. Meanwhile, there is a strong anticorrelation along $x$ for $|d_y| \leq 1$, which we attribute to the antibunching of individual holes in the same chain. We also compute the average

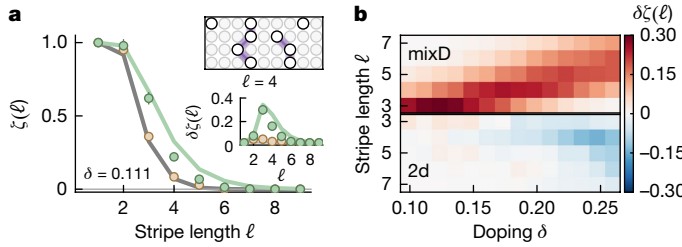

**Fig. 4 | Stripe-length histograms. a**, Normalized histogram of 'stripes' (as defined in the text) of at least length $\ell$ in mixD (green) and 2d (brown) systems at a doping of $\delta = 0.111$ compared with a random distribution of holes (grey line). We compare with a mean-field theory (see Supplementary Information) at $k_B T/t_x = 0.355$ (light-green line). The inset shows the difference $\delta\zeta(\ell)$ to the random distribution (see Supplementary Information). Error bars are estimated using bootstrapping and are smaller than the marker size if not visible. **b**, The full doping dependence of $\delta\zeta(\ell)$, in which the excess occurrences tend towards longer lengths with doping.

of the correlators over $d_y$ as $\overline{g^{(2)}}$ (top of Fig. 3b,c). This reveals a slightly positive signal at a distance of $d_x = 4$, qualitatively similar to Fig. 2. Although this signal is reminiscent of a charge density wave, future studies are required to confirm this hypothesis.

For further insights into the binding of larger structures, we consider the fully connected three-point hole–hole–hole correlator

$$C_{hhh}^{c}(\mathbf{d}^h, \mathbf{d}) = \frac{1}{\mathcal{N}_{\mathbf{d}^h,\mathbf{d}}} \sum_{\substack{\mathbf{i} \\ \mathbf{j}=\mathbf{i}+\mathbf{d}^h/2+\mathbf{d}}} \left( \frac{\langle \hat{n}_{\mathbf{i}}^h \hat{n}_{\mathbf{i}+\mathbf{d}^h}^h \hat{n}_{\mathbf{j}}^h \rangle - C_{disc}}{\langle \hat{n}_{\mathbf{i}}^h \rangle \langle \hat{n}_{\mathbf{i}+\mathbf{d}^h}^h \rangle \langle \hat{n}_{\mathbf{j}}^h \rangle} \right) \quad (4)$$

in which $C_{disc}$ removes all lower-order disconnected parts of the correlator (see Supplementary Information). We show the correlator for $\mathbf{d}^h = (0, 1)$ in Fig. 3d and find a positive signal at the closest distance along $y$, whereas all other distances are negative (along $x$) or vanish within the error bars (see Supplementary Information). This signal directly points to extended charge structures being favoured in excess of just individual hole pairs.

To provide further evidence for extended, fluctuating charge structures, we make use of the full information in our snapshots and count 'stripes'. To this end, we define a fully filled 'stripe' as a connected line of holes along $y$, for which the pairwise distance along $x$ between holes in neighbouring chains is at most 1 (see Fig. 4a, inset). We designate the length $\ell$ of this structure by the number of chains involved. We then consider the fraction $\zeta(\ell)$ of experimental realizations containing a 'stripe' of at least length $\ell$. In Fig. 4a, we compare the mixD case (green) with the 2d system (brown) and randomly distributed holes (grey line; see Supplementary Information) at a doping of $\delta = 0.111$ analysed on a subsystem of $9 \times 9$ sites. For the mixD case, we find an excess of events for large $\ell$, consistent with the tendency to form long fluctuating structures, whereas the results obtained for the standard 2d case are consistent with randomly distributed holes. Full numerical calculations are out of reach at our system size and temperature range, but a mean-field model of stripes shows quantitative agreement in the low-doping regime (see green lines in Fig. 4a and Supplementary Information). We next analyse the difference to the random distribution $\delta\zeta(\ell)$ as a function of doping (Fig. 4b). For all doping levels and lengths, this signal is positive in the mixD system, indicating the inclination of the system to form extended structures. The excess probability at longer lengths grows with doping as structures of increasing lengths form.

## Spin sector

The AFM correlations in the system and their interplay with charge delocalization are crucial for the formation of stripes and leads to characteristic signatures in the spin sector[13]. Most prominently, one

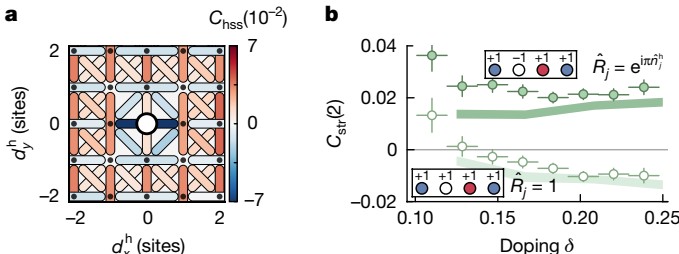

**Fig. 5 | Spin-sector analysis. a**, Hole–spin–spin correlation map. We show the bare correlator for diagonal and selected next-nearest spin bonds as a function of distance from the hole. The strongest signal is found in the sign change of the next-nearest-neighbour bond across the hole along $x$ pointing towards a domain wall in the local AFM pattern. Along $y$, the correlations keep their expected positive sign from the AFM pattern. **b**, Similarly, by considering the string spin correlator (dark green) and normal spin–spin correlator (light green) at distance $d = 2$ (see text), we observe a change in sign, consistent with the change in parity of the AFM pattern. Shaded regions are theory results on $L_x \times L_y = 8 \times 3$ at $k_B T/t_x = 0.3$. Error bars are estimated using bootstrapping and are smaller than the marker size if not visible.

expects a change in the parity of the AFM order in the presence of stripes, manifesting as incommensurate magnetism of the system and splitting of the peak at $(\pi, \pi)$ in the spin structure factor[5] (as also known in 1d systems[48]). Although our anisotropic and strongly interacting parameter regime is not favourable to investigate structure factors (see Supplementary Information), our microscopic resolution in both spin and charge sector allows us to evaluate real-space observables inaccessible in solid-state experiments. Most useful in this context are higher-order spin-charge correlators, such as the normalized, bare three-point hole–spin–spin correlator

$$C_{hss}(\mathbf{d}^s, \mathbf{d}^h) = \frac{1}{\mathcal{N}_{\mathbf{d}^s,\mathbf{d}^h}} \sum_{\substack{\mathbf{i} \\ \mathbf{j}=\mathbf{i}+\mathbf{d}^h-\mathbf{d}^s/2}} \frac{\langle \hat{n}_{\mathbf{i}}^h \hat{S}_{\mathbf{j}}^z \hat{S}_{\mathbf{j}+\mathbf{d}^s}^z \rangle}{\langle \hat{n}_{\mathbf{i}}^h \rangle \sigma(\hat{S}_{\mathbf{j}}^z) \sigma(\hat{S}_{\mathbf{j}+\mathbf{d}^s}^z)}, \quad (5)$$

in which $\mathbf{d}^s$ is the spin bond vector, $\mathbf{d}^h$ the distance of the bond from the dopant and we normalize by hole density $\langle \hat{n}^h \rangle$ and the spin standard deviation $\sigma(\hat{S}^z)$.

Previous studies have shown that, in square lattice 2d Fermi–Hubbard systems, a single mobile dopant will be surrounded by a dressing cloud of reduced spin correlations, forming a magnetic polaron[30–32]. In 1d systems, incommensurate magnetism leads to a change in the parity of the AFM pattern across impurities[49,50]. The same feature is predicted to prevail in stripe phases, making this correlator suited to revealing this specific feature in our data. We show the bare hole–spin–spin correlator as defined in equation (5) for the mixD system in Fig. 5a, in which specific spin bonds are shown for varying distances from a hole. We focus on the diagonal and next-nearest-neighbour correlators. The most prominent feature is the strongly negative correlation across the hole along $x$, which is consistent with a change in the parity of the local AFM pattern across a hole. Similarly, the diagonal bonds in the direct vicinity of the hole also become negative. This is another indication of fluctuations along $x$ within charge structures. Meanwhile, the $\mathbf{d}^s = (0, 2)$ correlations along $y$ are largely unaffected by the presence of a hole and retain their positive sign. The slightly negative (positive) $\mathbf{d}^s = (2, 0)$ ($\mathbf{d}^s = (1, 1)$) bond in the background further away from the dopant is a result of the overall doping level and vanishes in the connected correlator (see Supplementary Information).

Another way to explain the change in spin order across dopants is by using a spin-string correlator[49,51]. This spin–spin correlator has extra sign changes for every hole between two spins in the same chain and is defined as

$$C_{\text{str}}(d) = \frac{1}{\mathcal{N}_d} \sum_i \frac{\langle \hat{S}_i^z (\prod_{j=1}^{d-1} \hat{R}_{i+j}) \hat{S}_{i+d}^z \rangle - \langle \hat{S}_i^z \rangle \langle \hat{S}_{i+d}^z \rangle}{\sigma(\hat{S}_i^z)\sigma(\hat{S}_{i+d}^z)}, \qquad (6)$$

in which $\hat{R}_i = e^{i\pi\hat{n}_i^{\text{h}}}$. Note that, for $\hat{R}_i = \mathbb{1}$, the common spin–spin correlator is recovered. For systems with spin-charge separation, this correlator reveals a hidden spin structure in doped AFM systems[49,50]. The changes in the phase of the AFM pattern for stripe phases act in a similar fashion and can be revealed by measuring this string correlator along the direction perpendicular to the stripes (that is, along $x$). We show both the common spin–spin and the string correlator at distance $d = 2$ in Fig. 5b as a function of doping. We observe a change to a positive sign following the use of the string correlator that only varies weakly with doping, in agreement with theory predictions. These features can be directly related to the characteristic spin domain parity flips present in stripe phases. Note that we observe these features even without long-range AFM correlations—which are only expected at lower temperatures—because stripe-like structures already energetically favour such a local spin arrangement.

## Conclusion

We have realized a mixD Fermi–Hubbard model using ultracold atoms and found signatures of hole pairing and extended charge ordering in a temperature regime with short-ranged spin correlations, for which the collective behaviour of charges remains poorly understood. We detect effective hole attraction in density correlations and present further evidence for the onset of fluctuating individual stripes and their interplay with the magnetic background using real-space observables. Also, the spin environment is in qualitative agreement with the formation of an AFM domain wall across the dopants in both three-point and string correlators. We interpret these features as signatures for the formation of individual stripes as a precursor to the ordered stripe phase. The favourable energy scales of the mixD setting pave the way for quantum simulators to study this collective phase, including the precise periodicity, fluctuations and filling[6], and thereby provide valuable comparisons with recent results in theoretical calculations[17,52]. The direct connection between mixD and 2d systems provides a possible method to study the adiabatic preparation of stripes using mixD couplings. Through the mapping to attractive interactions[53], new insights into the stripe phase also directly relate to the exotic Fulde–Ferrell–Larkin–Ovchinnikov phase[54]. Furthermore, direct extensions to bilayer mixD systems connect our work to recently discovered high-$T_c$ compounds, for which the mixed dimensionality seems essential for the emergence of a superconducting phase at around 80 K in bilayer nickelates[7,8,10].

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

# Methods

### Experimental sequence

We prepare a spin-balanced sample of ultracold $^6$Li atoms in the two lowest hyperfine states $|F = 1/2, m_F = \pm 1/2\rangle$ in a single layer of an optical lattice following our previous work[36,55]. After magnetic evaporation, we load from a crossed dipole trap into a box potential (surrounded by a reservoir with approximately $h \times 2$ kHz higher chemical potential) projected with a digital mirror device (see Extended Data Fig. 1a). From this, we load into optical lattices along $x$ and $y$ with $a_x = 1.11$ μm and $a_y = 1.14$ μm. Their depths in the following are given in units of their respective lattice recoil $E_R = h^2/(8Ma^2)$, in which $M$ is the atomic mass.

To prepare the mixD system described in the main text without introducing large density inhomogeneities, we cannot directly load into the final lattice configuration but instead follow a procedure similar to that in ref. 36 (see Extended Data Fig. 1c). We first load into decoupled 1d chains along $x$ by exponentially ramping to $V_x = 3E_R$, $V_y = 35E_R$ and a scattering length of $1,160a_B$, corresponding to our final on-site interactions of $U = h \times 4.4(1)$ kHz within 200 ms. At this point, we turn on a superlattice along $y$ ($a_y^{SL} = 2a_y = 2.28$ μm) to a depth of $V_y^{SL} = 2E_R$ within 1 ms. By tuning the relative phase between the lattice and the superlattice to the fully staggered configuration, we ensure that the spin couplings remain the same in even and odd bonds along $y$. This staggering creates a potential offset of $\Delta = 0.65(5)U$ between neighbouring sites to suppress tunnelling along $y$. We then slowly restore coupling along $y$ by ramping the lattices in 56 ms to their final depths of $V_x = 9E_R$ and $V_y = 7E_R$. We make sure to keep the interactions constant during this second ramp by adjusting the scattering length accordingly, leading to a final scattering length of $1,293a_B$. For any 2d system comparison, we perform the same ramps without turning on the $y$ superlattice. For more details on our superlattice design, see ref. 56. For detection, we freeze out the system by ramping to $V_{x/y} = 43.5E_R$ within 1.5 ms and perform spin-resolved single-site detection as described in ref. 36.

The resulting system can be accurately described by a Fermi–Hubbard-type model with parameters $(t, U, \Delta)$, which can be mapped onto the $t$–$J$ model of equation (1). For all settings, we have tunnelling $t_x = h \times 163(10)$ Hz, interactions $U = h \times 4.4(1)$ kHz (thus $U/t_x = 27(2)$) and superexchange $J_x = h \times 24(4)$ Hz. For the 2d system, we have $t'_y = h \times 253(13)$ Hz, however for $\Delta \neq 0U$, the effective coupling $t_y$ is negligible. The superexchange coupling $J_y$ is nonzero in both cases with $J'_y = h \times 58(7)$ Hz for the 2d system and $J_y = h \times 104(23)$ Hz for $\Delta = 0.65(5)U$. Owing to the strongly anisotropic spin couplings and large $U/t_x$, the spin correlations are not sufficiently long-ranged to expect any signal in the spin structure factor.

We estimate the temperature of our system using the spin correlations $C_{ss}(\mathbf{d}) = \frac{1}{\mathcal{N}_{\mathbf{d}}} \sum_{\mathbf{i}} \frac{\langle \hat{S}_{\mathbf{i}}^z \hat{S}_{\mathbf{i+d}}^z \rangle - \langle \hat{S}_{\mathbf{i}}^z \rangle \langle \hat{S}_{\mathbf{i+d}}^z \rangle}{\sigma(\hat{S}_{\mathbf{i}}^z)\sigma(\hat{S}_{\mathbf{i+d}}^z)}$ as a function of doping and compare this to matrix product states (MPS) calculations in Extended Data Fig. 2. We fit the individual doping bins to the numerical data and extract their respective temperature. As the short $y$ direction may be subject to finite size effects in the DMRG calculations, we determine individual temperatures along $x$ and $y$. By extracting temperatures per doping level, we estimate a temperature of $k_B T/t_x \approx 0.3(1)$ and $k_B T/t_x \approx 0.4(1)$ from the correlations along $x$ and $y$, respectively. Owing to the doping and high interactions, spin correlations only extend up to approximately three sites along $y$ and are shorter along $x$. For that reason, fully correlated domain walls cannot form, making structure factors particularly challenging to investigate.

### Offset phase calibration

To calibrate the detuning $\Delta$, we first need to precisely determine the relative phase between the lattice and the superlattice. For this, we load a dilute cloud into a system of decoupled double wells along $y$, in which $V_x = 40E_R$, $V_y = 8E_R$ and $V_y^{SL} = 21E_R$, leading to an intrawell coupling of $t_y(\phi = 0) = h \times 724(80)$ Hz. We vary the phase between the lattice and

the superlattice and measure the normalized imbalance, that is, the difference in occupation between the different parts of the double well, normalized by their summed occupation (see Extended Data Fig. 3). When we prepare symmetric double wells, the imbalance approaches zero. However, when we tune away from this configuration, we reach an imbalance of $\pm 1$ within less than 50 mrad. This sharp transition indicates a high degree of stability and homogeneity of the relative superlattice phase within the system (see also ref. 56 for more details). For the measurements presented here, we then work at a phase of $\phi = \pi/2$, at which the offset between neighbouring lattice sites is highest for a given lattice depth and the interwell and intrawell couplings are identical.

We confirm the energy scales associated with a given potential offset by comparing it with our interaction energy. We prepare the system at the lattice parameters stated in the previous section and a phase of $\phi = \pi/2$ and vary the depth of the superlattice (that is, the potential offset) in a slightly hole doped system (see Extended Data Fig. 3c,d). For offsets smaller than the bandwidth, tunnelling between sites is not yet suppressed and we create a strong imbalance. On the other hand, for large offsets around the interaction energy, we enable resonant tunnelling between sites whenever both are occupied, thus creating both an imbalance and doublons within the system. The observed scales are consistent with band-structure calculations based on our lattice parameters. Between these two regimes, the imbalance approaches zero. The maximum in imbalance around $U/2$ can be explained by a second-order process in which a doublon in a lower chain breaks into two atoms in the two adjacent chains (see also ref. 56). To avoid this effect and the associated extra holes and doublons, we perform our experiments at an offset slightly above $U/2$. The resulting mixD system has a small residual normalized density imbalance (as can also be seen in Extended Data Fig. 1b) of about 0.037. This does not affect the validity of the Hubbard Hamiltonian of equation (1) (as the couplings are unchanged) but only leads to a slightly worse doping resolution.

### Data statistics, doping histograms

In total, we collect 11,675 experimental realizations. Of these, 1,254 were taken in a 2d system with $\Delta = 0U$, the remaining 10,421 with $\Delta = 0.65(5)U$. Within these measurements, we slightly vary the doping level (as well as the natural fluctuations inherent to our preparation scheme) which yields a range of 10–30% hole doping. To ensure that there is no overall magnetization $M^z = \sum_i \langle \hat{S}_i^z \rangle$ within the system, we check the distribution of magnetization normalized by the system size, which is centred around zero and shows a width below shot noise (see Extended Data Fig. 4).

### Connected correlators and offsets

**Full connected correlator expressions.** We present a variety of correlators to characterize the spin and charge order in our system. We here distinguish between bare, 'partially connected' and fully connected correlators. Although the bare correlator does not subtract anything, the fully connected correlator subtracts all possible lower-order contributions between all of its constituents, for example, for a two-point correlator, it removes the product of the mean operator values, whereas for a three-point correlator, it also removes all combinations of two-point correlators. Meanwhile, partially connected correlators only subtract some specific lower-order correlators: in this case, as we consider pairs as new objects, we do not subtract any correlations arising from the individual holes in the pair.

All of these different types of correlator are then helpful to extract slightly different information about the system. Although fully connected correlators are especially useful to extract small signals in higher-order correlators dominated by lower-order contributions, the bare correlator may be more interesting when higher-order correlations are actually larger than lower-order correlators.

For this reason, as well as the partially connected pair–hole and pair–pair correlator (equation (3)) and the bare hole–spin–spin correlator (equation (5)), we used the fully connected hole–hole–hole correlator defined as

$$
\begin{aligned}
C_{hhh}^{c}(\mathbf{d}^{h},\mathbf{d}) = \frac{1}{\mathcal{N}_{\mathbf{d}^{h},\mathbf{d}}} \sum_{\substack{\mathbf{i} \\ \mathbf{j}=\mathbf{i}+\mathbf{d}^{h}/2+\mathbf{d}}} & \frac{1}{\langle \hat{n}_{\mathbf{i}}^{h}\rangle \langle \hat{n}_{\mathbf{i}+\mathbf{d}^{h}}^{h}\rangle \langle \hat{n}_{\mathbf{j}}^{h}\rangle} \times (\langle \hat{n}_{\mathbf{i}}^{h}\hat{n}_{\mathbf{i}+\mathbf{d}^{h}}^{h}\hat{n}_{\mathbf{j}}^{h}\rangle \\
& - \langle \hat{n}_{\mathbf{i}}^{h}\hat{n}_{\mathbf{i}+\mathbf{d}^{h}}^{h}\rangle \langle \hat{n}_{\mathbf{j}}^{h}\rangle - \langle \hat{n}_{\mathbf{i}}^{h}\rangle \langle \hat{n}_{\mathbf{i}+\mathbf{d}^{h}}^{h}\hat{n}_{\mathbf{j}}^{h}\rangle \\
& - \langle \hat{n}_{\mathbf{i}}^{h}\hat{n}_{\mathbf{j}}^{h}\rangle \langle \hat{n}_{\mathbf{i}+\mathbf{d}^{h}}^{h}\rangle + 2\langle \hat{n}_{\mathbf{i}}^{h}\rangle \langle \hat{n}_{\mathbf{i}+\mathbf{d}^{h}}^{h}\rangle \langle \hat{n}_{\mathbf{j}}^{h}\rangle).
\end{aligned}
\tag{7}
$$

Similarly, we can define a connected hole–spin–spin correlator as

$$
\begin{aligned}
C_{hss}^{c}(\mathbf{d}^{s},\mathbf{d}^{h}) = \frac{1}{\mathcal{N}_{\mathbf{d}^{s},\mathbf{d}^{h}}} \sum_{\substack{\mathbf{i} \\ \mathbf{k}-\mathbf{j}=\mathbf{d}^{s}, \\ (\mathbf{k}+\mathbf{j})/2-\mathbf{i}=\mathbf{d}^{h}}} & \frac{1}{\langle \hat{n}_{\mathbf{i}}^{h}\rangle \sigma(\hat{S}_{\mathbf{j}}^{z})\sigma(\hat{S}_{\mathbf{k}}^{z})} \times (\langle \hat{n}_{\mathbf{i}}^{h}\hat{S}_{\mathbf{j}}^{z}\hat{S}_{\mathbf{k}}^{z}\rangle \\
& - \langle \hat{n}_{\mathbf{i}}^{h}\rangle \langle \hat{S}_{\mathbf{j}}^{z}\hat{S}_{\mathbf{k}}^{z}\rangle - \langle \hat{n}_{\mathbf{i}}^{h}\hat{S}_{\mathbf{j}}^{z}\rangle \langle \hat{S}_{\mathbf{k}}^{z}\rangle \\
& - \langle \hat{n}_{\mathbf{i}}^{h}\hat{S}_{\mathbf{k}}^{z}\rangle \langle \hat{S}_{\mathbf{j}}^{z}\rangle + 2\langle \hat{n}_{\mathbf{i}}^{h}\rangle \langle \hat{S}_{\mathbf{j}}^{z}\rangle \langle \hat{S}_{\mathbf{k}}^{z}\rangle).
\end{aligned}
\tag{8}
$$

For this connected correlator, we observe the same main features also shown in Fig. 5a with a dominant negative bond across the hole (see Extended Data Fig. 5). This signal is strong enough to dominate over the AFM background, changing the correlator sign even in the bare correlator shown in Fig. 5. Meanwhile, the positive diagonal and next-nearest-neighbour bonds along $y$ far from the hole shown in Fig. 5a now vanish, as they are not related to the presence of a hole but just stem from the AFM background. We compare this to DMRG calculations with $L_y = 4$, $\delta = 0.125$ and $k_B T/t_x = 0.4$, which shows the same main features of strong anticorrelations across the dopant and at the diagonals in the immediate vicinity.

**Offset correction.** As well as the subtraction of the disconnected part, we also introduce an offset correction $o_\delta$ on the hole–hole correlator. This correction arises owing to the doping fluctuations in our finite-sized system. For each realization, we prepare a system with random but fixed total atom number and magnetization (see Extended Data Fig. 4). The calculated correlations in a finite system then obey a sum rule depending on the particle number and variance.

We start by considering $N$ fermions on $V$ sites with density $n = N/V$. The local two-point correlator $\Gamma(i,j) = \frac{\langle \hat{n}_i \hat{n}_j \rangle}{n_i n_j} - 1$ (with $n_i = \langle \hat{n}_i \rangle$) after summing over all possible pairs of sites $i,j$ can be expressed as

$$
\sum_{i,j} \Gamma(i,j) = \sum_{i,j} \left( \frac{\langle \hat{n}_i \hat{n}_j \rangle}{n_i n_j} - 1 \right) \approx \left( \frac{\langle \hat{N}^2 \rangle}{n^2} - V^2 \right),
\tag{9}
$$

in which we used $\hat{N}^2 = \sum_{i,j} \hat{n}_i \hat{n}_j$ and $n_i \approx n_j \approx n$ in our homogeneous system. This we can relate to the variance as

$$
\frac{1}{V^2} \sum_{i,j} \Gamma(i,j) = \frac{\mathrm{Var}(\hat{N})}{N^2}.
\tag{10}
$$

If we now separate the on-site fluctuations and use fermionic statistics in which $\hat{n}^2 = \hat{n}$ (and thus $\Gamma(i,i) = \frac{1}{n} - 1$), we obtain

$$
\frac{1}{V^2} \sum_{i \neq j} \Gamma(i,j) = \frac{\mathrm{Var}(\hat{N}) - N(1-n)}{N^2}.
\tag{11}
$$

Unless the global fluctuations of $N$ are also fermionic fluctuations (that is, multinomial, in which $\mathrm{Var}(\hat{N}) = N(1-n)$), the sum rule in equation (11) leads to a nonzero value of $\Gamma(i,j)$ for $i \neq j$ even at $T = \infty$. Note that typically $\mathrm{Var}(\hat{N}) \propto N$ or less, such that equation (11) is a $1/N$ correction, which vanishes in the thermodynamic limit.

Identifying $\hat{n} \equiv \hat{n}^h$, we use this result in the calculation of the hole–hole correlations in Fig. 2 and thus define the offset $o_\delta$ through

$$
o_\delta = \frac{\mathrm{Var}(\hat{N}^h) - N^h(1-n^h)}{(N^h)^2},
\tag{12}
$$

and the corrected correlation as equation (2)

$$
g_{hh}^{(2)}(\mathbf{d}) - 1 = \frac{1}{\mathcal{N}_{\mathbf{d}}} \sum_{\mathbf{i}} \left( \frac{\langle \hat{n}_{\mathbf{i}}^{h}\hat{n}_{\mathbf{i}+\mathbf{d}}^{h}\rangle}{\langle \hat{n}_{\mathbf{i}}^{h}\rangle \langle \hat{n}_{\mathbf{i}+\mathbf{d}}^{h}\rangle} - 1 - o_\delta \right),
\tag{13}
$$

with $\mathcal{N}_{\mathbf{d}} = \sum_{\mathbf{i},\mathbf{j}} \delta_{\mathbf{j},\mathbf{i}+\mathbf{d}}$ and Kronecker delta $\delta_{i,j}$. Most importantly, the doping-dependent offset we apply is global on all distances. Therefore, we can understand this offset as a global $-1/N$ correction for a fixed number of particles in the system, whereas for exceedingly large global fluctuations, positive offsets can occur.

This offset correction $o_\delta$ only plays a role when selecting specific doping levels in a finite-sized system such that the total atom number is almost fixed ($\mathrm{Var}(\hat{N}) \rightarrow 0$) and thereby leads to strong global offsets, that we hereby compensate (see Extended Data Fig. 6a). We show in Extended Data Fig. 6b the offset as a function of doping together with the nearest-neighbour hole–hole correlator values with and without applied offset. As indicated by the dashed lines, the offset without selection on a density bin is negligible. For this reason, we do not apply any corrections in Fig. 3.

**Correlator from theory.** When comparing the absolute values of hole–hole correlations with simulations, care needs to be taken because of the differences in doping, fluctuations and boundary conditions. All calculations are performed with open boundary conditions along $x$ and $y$. Meanwhile, the potential at the edges in the experiment has a finite width, which means that the exact position of any charge feature will be fluctuating and therefore be washed out. As a result, we detect signals in $g_{hh}^{(2)}$ but not in the density, in contrast to theory, in which stripes appear as density features[36]. When using connected correlators on theory data, this will lead to reduced correlations. To analyse numerical results, we hence use the slightly modified correlator $\widetilde{g}_{hh}^{(2)}(\mathbf{d})$ defined as

$$
\widetilde{g}_{hh}^{(2)}(\mathbf{d}) - 1 = \frac{1}{\mathcal{N}_{\mathbf{d}}} \sum_{\mathbf{i}} \left( \frac{\langle \hat{n}_{\mathbf{i}}^{h}\hat{n}_{\mathbf{i}+\mathbf{d}}^{h}\rangle}{n^h n^h} - 1 \right)
\tag{14}
$$

in which, compared with equation (2), we replace the normalization by the local densities with the global doping level $n^h$. This effectively assumes that the density is homogeneous throughout the system instead of bunched at the centre, allowing for easier comparison with the experiment.

**Statistical significance in correlation maps.** The correlation maps shown in Figs. 2 and 3 do not give any indication of which data points in the map are statistically significant or fall below the noise floor of the measurement. To address this, we show in Extended Data Fig. 6c–f the same maps as in Figs. 2 and 3 for which we now set all distances with signals compatible with zero (that is, the signal being less than $1\sigma$ away from zero) to grey. All features mentioned in the main text are still clearly visible.

## Further coupling terms in the mixD Fermi–Hubbard model

In the experiment, we realize a 2d Fermi–Hubbard model with anisotropic tunnel couplings and energy offset on every second site along $y$. In the limit of strong interactions $U \gg t_x, t_y$ used here, this is commonly mapped onto the $t$–$J$ model. However, this approximation neglects higher-order terms that can arise in the expansion, including a crucial second-order hopping term. Although nearest-neighbour hopping is suppressed owing to the potential offset, next-nearest-neighbour

hopping remains resonant in a staggered potential. We experimentally confirmed the presence of this term and its scaling $\tilde{t}_y = t_y'' + \frac{t_y^2}{\Delta}$ with direct next-nearest-neighbour tunnelling $t_y''$ (which is, however, negligible for our parameters) by performing single-particle quantum walks[56]. This simple expression neglects interaction effects with atoms in the intermediate lattice site. For $\Delta = 0.65(5)U$, this means that $\tilde{t}_y \approx \frac{1.54 t_y^2}{U}$. This could, in principle, disfavour stripe formation, as the weak Pauli repulsion associated with $\tilde{t}_y$ could inhibit pairs at distance 2 such that only $d_y = 1$ hole pairs would form. In this experiment, the contribution can mostly be neglected as the principal energy scale is given by $J_y \approx 3\tilde{t}_y$, which dominates in our parameter regime over $\tilde{t}_y$.

## Stripe-length random data generation

To interpret the stripe-length results of Fig. 4, we compare with random hole distributions with different short-ranged correlations. We first simply randomly sample holes on 9 × 9 sites (see Extended Data Fig. 7a), in which we observe strong positive signals in the mixD case and negative signals for the 2d case. However, the strong Pauli repulsion along $x$ might have an influence on this signal. For this reason, we randomly sampled holes for which we included, in Fig. 4, the experimentally obtained anticorrelations along $x$ (see Fig. 2). Finally, we compare with randomly placed pairs along $y$ within the system in Extended Data Fig. 7b, exhibiting similar features. Thus, we conclude that the observed main qualitative features are relatively insensitive to the exact details of the randomly generated data and that we see a genuine stripe signal that cannot be explained by random or short-ranged correlated holes.

## Numerical simulations of the mixD $t$–$J$ model

We simulate the mixD $t$–$J$ model,

$$
\hat{\mathcal{H}} = \hat{\mathcal{P}} \left( -t_x \sum_{\langle \mathbf{i},\mathbf{j} \rangle_x} \sum_{\sigma = \uparrow, \downarrow} \hat{c}_{\mathbf{i},\sigma}^{\dagger} \hat{c}_{\mathbf{j},\sigma} + \text{h.c.} \right) \hat{\mathcal{P}} + J_x \sum_{\langle \mathbf{i},\mathbf{j} \rangle_x} \left( \mathbf{S}_\mathbf{i} \cdot \mathbf{S}_\mathbf{j} - \frac{\hat{n}_\mathbf{i} \hat{n}_\mathbf{j}}{4} \right)
$$
$$
+ J_y \sum_{\langle \mathbf{i},\mathbf{j} \rangle_y} \left( \mathbf{S}_\mathbf{i} \cdot \mathbf{S}_\mathbf{j} - \frac{\hat{n}_\mathbf{i} \hat{n}_\mathbf{j}}{4} \right),
\tag{15}
$$

(see equation (1)) for $J_y/t_x = 0.5$ and $J_x/J_y = 0.3$ at finite temperature using MPS through mixed state purification schemes[57–59]. In particular, we expand the system by introducing one auxiliary site for each physical site, which allows for showing mixed physical states as pure states on an enlarged Hilbert space. A pure state in the enlarged system at finite temperature is calculated by evolving the maximally entangled, infinite temperature state $|\Psi(\beta = 0)\rangle$ in imaginary time under the physical Hamiltonian, $|\Psi(\tau)\rangle = e^{-\tau \hat{\mathcal{H}}} |\Psi(\beta = 0)\rangle$, in which $\tau = \beta/2$, with $\beta$ the inverse temperature. Thermal expectation values $\langle \hat{O} \rangle_\text{T}$ in the physical subset are computed by tracing out the auxiliary degrees of freedom, that is,

$$
\langle \hat{O} \rangle_\text{T} = \frac{\langle \Psi(\beta) | \hat{O} | \Psi(\beta) \rangle}{\langle \Psi(\beta) | \Psi(\beta) \rangle}.
\tag{16}
$$

During the imaginary time evolution, we conserve the particle number in each row $N_\ell$, $\ell = 1, \ldots, L_y$, the total particle number in the auxiliary system $N_\text{aux.}^\text{tot}$, and the total spin $S_\text{phys.+aux.}^{z,\text{tot}}$ (the latter allowing for thermal fluctuations of the total magnetization in the physical system). This results in a total of $L_y + 2$ symmetries used by the DMRG implementation, leading to marked speed-ups over a single global $U(1)$ conservation in the overall physical system[11].

The maximally entangled state needed as a starting point of the imaginary time evolution, $|\Psi(\beta = 0)\rangle$, is generated using specifically tailored entangler Hamiltonians[11,60]. Because these states (being projected product states) are of low bond dimension ($\chi(\tau = 0) \approx \mathcal{O}(100)$), local approximations of the Hamiltonian and subsequent exponentiation will suffer from large projection errors. Hence, we start by using

global methods for a single step in imaginary time, after which the entanglement in the system (and the bond dimension of the thermal MPS) has sufficiently increased to switch to local methods.

Owing to the mapping of the (enlarged) 2d system to a 1d chain, the bond dimension required for a fixed accuracy scales exponentially with linear system size in the $y$ direction. For doping scans, we limit the system size to $L_x \times L_y = 8 \times 3$ with open boundaries and hole configurations $N_\ell = 1, 2, 3$ for each $\ell = 1, 2, 3$. For a single hole per chain, we simulate systems up to $L_y = 4$. As this mixD model suffers from the fermion sign problem, these limited system sizes are still state of the art for numerical calculations while mostly allowing general qualitative comparison with the much larger experimental system. Larger system sizes have only been achieved at zero temperature, which is numerically much easier to realize in DMRG. We furthermore checked that our temperature estimations (Extended Data Fig. 2) are not affected by the finite size effects of the DMRG calculation by comparing spin correlations for $L_y = 3$ and $L_y = 4$ at $\delta = 0.125$ and finding very similar values.

In particular, we evolve $|\Psi(\beta = 0)\rangle$ using global Krylov schemes by a single step $t_x \Delta \tau = 0.01$. Weight cut-offs are set to $10^{-10}$, expanding the bond dimension to $\chi(\tau = \Delta \tau) \approx \mathcal{O}(1,000)$. From here on, we switch to the local two-site time-dependent variational principle (TDVP) method[59] with time steps of $t_x \Delta \tau = 0.03$, weight and truncation cut-offs of $10^{-10}$ and $10^{-12}$, respectively, and cutting edge maximum bond dimensions of $\chi_\text{TDVP} = 30,000$. We evolve the system to $\tau t_x = 2.0$, corresponding to a temperature of $k_\text{B} T / t_x = 0.25$.

Spin–spin correlations, as well as hole distributions in each leg, are exemplarily shown in Extended Data Fig. 8a for $k_\text{B} T / t_x \approx 0.4$ for a system of size $L_x \times L_y = 8 \times 4$ with periodic boundaries along the short direction and $N_\ell = 1$ for all $\ell = 1, \ldots, 4$. At the centre of the chains, at which the hole density peaks, an AFM domain wall forms, signalling the formation of a single, fully filled stripe. For a higher doping of $\delta = 0.25$ ($L_y = 3$, open boundaries), we show the hole density as well as hole–hole correlations in Extended Data Fig. 8b,c, in which the two separate stripes are visible. Results as a function of temperature are shown in Extended Data Fig. 8d for $d_y = 1$ and $d_y = 2$.

## Effective descriptions of stripes in the mixD $t$–$J$ model

**Mean-field theory.** In this section, we present a mean-field theory for the stripe phase in the mixD $t$–$J$ model. We focus on describing an individual stripe in the $y$ direction with exactly one hole per chain, bound by the magnetically mediated confining potentials. In particular, we neglect the interaction between several stripes at positions $\mathbf{i}_1$ and $\mathbf{i}_2$, that is, we focus on the low-doping regime. To illustrate the concept, we first consider a mean-field description of the ground state, before generalizing to finite temperature.

For $t_x \gg J_x, J_y$, quantum correlations between strongly fluctuating holes and spins in squeezed space (defined in refs. 51,61) can be neglected[62–66]. Hence, we make the ansatz

$$
|\psi\rangle = |\psi\rangle_\text{sq} \otimes |\psi\rangle_\text{c},
\tag{17}
$$

in which $|\psi\rangle_\text{sq}$ is the spin state of the undoped Heisenberg model in squeezed space and $|\psi\rangle_\text{c}$ is the chargon wavefunction. Our starting point for the description of the single stripe is the variational Gutzwiller wavefunction, given by

$$
|\psi\rangle_\text{c} = \bigotimes_{y=-\infty}^{\infty} |\phi^{(0)}\rangle_y,
\tag{18}
$$

that is, we describe the charge sector by the product of identical single-leg wavefunctions $|\phi^{(0)}\rangle_y$ in chain $y$. Assuming that the stripe is centred around $x = 0$, we express $|\phi^{(0)}\rangle_y$ within the string basis,

$$
|\phi^{(0)}\rangle_y = \sum_{\Sigma=-\infty}^{\infty} \phi_\Sigma^{(0)} |y, \Sigma\rangle,
\tag{19}
$$

in which $\Sigma$ can be understood as the length of the string measured relative to the centre of the stripe.

Within this variational ansatz, coefficients $\phi_\Sigma^{(0)}$ can be found by minimizing the energy of the trial state, $\langle\hat{\mathcal{H}}\rangle_0 = \langle\psi|\hat{\mathcal{H}}|\psi\rangle = (\langle\psi|_{sq} \otimes \langle\psi|_c)\hat{\mathcal{H}}(|\psi\rangle_{sq} \otimes |\psi\rangle_c)$,

$$\frac{\langle\hat{\mathcal{H}}\rangle_0}{L_y} = \frac{E_0}{L_y} - t_x \sum_\Sigma (\phi_{\Sigma+1}^{(0)*}\phi_\Sigma^{(0)} + \text{c.c.}) + \sum_{\Sigma,\Sigma'} |\phi_\Sigma^{(0)}|^2 |\phi_{\Sigma'}^{(0)}|^2 V_{\text{pot}}(\Sigma - \Sigma'). \quad (20)$$

Here $V_{\text{pot}}(\Sigma)$ is the interchain potential defined by the potential energy of two holes in neighbouring chains separated by the string $\Sigma$,

$$V_{\text{pot}}(\Sigma) = J_y[(|\Sigma| - 1 + \delta_{\Sigma,0})C_2 - (|\Sigma| + 1)C_1^y], \quad (21)$$

in which $C_1^\mu = \langle\psi_s|\hat{\mathbf{S}}_i \cdot \hat{\mathbf{S}}_{i+\mathbf{e}_\mu}|\psi_s\rangle$, $\mu = x, y$ are nearest neighbours and $C_2 = \langle\psi_s|\hat{\mathbf{S}}_i \cdot \hat{\mathbf{S}}_{i+\mathbf{e}_x+\mathbf{e}_y}|\psi_s\rangle$ are diagonal spin–spin correlations in the undoped Heisenberg model in the ground state. Note that there are also intrachain contributions, which, however, are constant and only lead to a trivial energy shift on top of the Heisenberg ground state energy $E_0$ (see Extended Data Fig. 9a).

By averaging over the upper and lower chains for a given leg, we can reformulate the variational problem, equation (20), as a self-consistent ground state search of the mean-field Hamiltonian per chain,

$$\hat{\mathcal{H}}_{\text{MF}} = \frac{E_0}{L_y} - t_x \sum_{\Sigma,\Sigma'} [\hat{h}_\Sigma^\dagger \hat{h}_\Sigma + \text{h.c.}] + \sum_\Sigma \hat{h}_\Sigma^\dagger \hat{h}_\Sigma V_{\text{eff}}(\Sigma), \quad (22)$$

in which $\hat{h}_\Sigma^\dagger|y, 0\rangle = |y, \Sigma\rangle$ and

$$V_{\text{eff}}(\Sigma) = 2 \sum_{\Sigma'} |\phi_{\Sigma'}^{(0)}|^2 V_{\text{pot}}(\Sigma' - \Sigma). \quad (23)$$

Note the factor of 2 in the potential energy, arising from energy contributions between chains $y \pm 1$ with chain $y$. When considering the total energy of the variational wavefunction, equation (20), however, there is no extra factor to not double count interchain energy contributions.

In practice, we set a maximal cut-off for the string length, here chosen as $|\Sigma_{\max}| \approx 15$. By exact diagonalization and self-consistently solving equation (22), the string-length distribution $|\phi_\Sigma^{(0)}|^2$ within the mean-field picture can be calculated.

At finite temperature, we generalize the ansatz to a product of density matrices,

$$\hat{\rho} = \hat{\rho}_{sq} \otimes \left( \bigotimes_{y=-\infty}^{\infty} \hat{\rho}_{\text{MF}}^{(0)} \right), \quad (24)$$

in which

$$\hat{\rho}_{\text{MF}}^{(0)} = \frac{1}{Z} e^{-\beta\hat{\mathcal{H}}_{\text{MF}}(\hat{\rho}_{\text{MF}}^{(0)}, T)} \quad (25)$$

defines the self-consistency equation through

$$\frac{\hat{\mathcal{H}}_{\text{MF}}(\hat{\rho}_{\text{MF}}^{(0)})}{L_y} = \frac{E_0}{L_y} - t_x \sum_{\Sigma,\Sigma'} [\hat{h}_\Sigma^\dagger \hat{h}_\Sigma + \text{h.c.}]$$
$$+ \sum_\Sigma \hat{h}_\Sigma^\dagger \hat{h}_\Sigma V_{\text{eff}}(\Sigma; \hat{\rho}_{\text{MF}}^{(0)}, T), \quad (26)$$

$$V_{\text{eff}}(\Sigma; \hat{\rho}_{\text{MF}}^{(0)}, T) = 2 \sum_{\Sigma'} \langle\Sigma|\hat{\rho}_{\text{MF}}^{(0)}|\Sigma\rangle V_{\text{pot}}(\Sigma' - \Sigma; T).$$

Here $C_1^\mu(T) = \langle\hat{\mathbf{S}}_i \cdot \hat{\mathbf{S}}_{i+\mathbf{e}_\mu}\rangle_T$, $\mu = x, y$ and $C_2(T) = \langle\hat{\mathbf{S}}_i \cdot \hat{\mathbf{S}}_{i+\mathbf{e}_x+\mathbf{e}_y}\rangle_T$ entering $V_{\text{pot}}$ in equation (21) are thermally averaged two-point correlators of the 2d Heisenberg model. Given the self-consistent solution of $\hat{\rho}_{\text{MF}}^{(0)}$, the mean-field string-length distribution is determined by the diagonal elements of $\hat{\rho}_{\text{MF}}^{(0)}$, that is, $p_\Sigma = \langle\Sigma|\hat{\rho}_{\text{MF}}^{(0)}|\Sigma\rangle$.

We use finite-temperature DMRG methods (see previous section) to calculate thermally averaged nearest-neighbour and diagonal correlations of the undoped Heisenberg model with $J_x/J_y = 0.3$ on a $L_x \times L_y = 12 \times 4$ lattice with periodic boundaries along $y$; see Extended Data Fig. 9b. Results for the corresponding mean-field estimates of the string-length distributions in the stripe phase are shown in Extended Data Fig. 9c for $t_x/J_y = 2$ and temperatures $k_BT/t_x = [0.2, 0.625]$.

Using the mean-field theory string-length distributions, we sample snapshots and compare the resulting stripe-length distributions to the experiment (see Extended Data Fig. 9f). At the expected temperature of $k_BT/t_x \approx 0.3$, the effective description matches the experiment rather well, with only a slight overestimation of the order in the mean-field description.

**Müller–Hartmann–Zittartz estimate.** To make further comparisons with statistical models, we reduce the mixD system to a 1d, purely classical model of fluctuating holes bound together by the effective potential $V_{\text{pot}}$ (equation (21); Müller–Hartmann–Zittartz (MHZ) approach). Denoting with $x_\ell$ the $x$ position of the doped hole in leg $\ell$ (we again consider one hole per chain, that is, a single fluctuating domain wall), the effective Hamiltonian (excluding quantum fluctuations from the hopping of the holes) for a system of size $(L_x + 1) \times (L_y + 1)$ reads

$$\hat{\mathcal{H}}_{\text{MHZ}} = \sum_{\ell=1}^{L_y} V_{\text{pot}}(|x_\ell - x_{\ell+1}|; T), \quad (27)$$

in which again the temperature-dependent correlators $C_1^\mu(T) = \langle\hat{\mathbf{S}}_i \cdot \hat{\mathbf{S}}_{i+\mathbf{e}_\mu}\rangle_T$, $\mu = x, y$ and $C_2(T) = \langle\hat{\mathbf{S}}_i \cdot \hat{\mathbf{S}}_{i+\mathbf{e}_x+\mathbf{e}_y}\rangle_T$ enter the effective potential $V_{\text{pot}}(|x_\ell - x_{\ell+1}|; T)$ in equation (21).

The partition function, $Z$, decouples when being expressed solely by distances $d_\ell = x_\ell - x_{\ell+1}$,

$$Z = \sum_{\{x_\ell\}} \prod_{\ell=1}^{L_y} \exp[-\beta V_{\text{pot}}(|x_\ell - x_{\ell+1}|; T)]$$
$$= \sum_{d_1=-L_x}^{L_x} \cdots \sum_{d_{L_y}=-L_x}^{L_x} \prod_{\ell=1}^{L_y} \exp[-\beta V_{\text{pot}}(|d_\ell|; T)] \quad (28)$$
$$= \left[ \sum_{d=-L_x}^{L_x} \exp[-\beta V_{\text{pot}}(|d|; T)] \right]^{L_y} = [Z_1]^{L_y}.$$

The probability of finding two adjacent holes at distance $d$ in chains $\ell, \ell+1$ is given by

$$p(d) = \exp[-\beta V_{\text{pot}}(|d|; T)]/Z_1, \quad (29)$$

shown for various temperatures $k_BT/J_y$ in Extended Data Fig. 9d.

Fixing the first hole in the centre and sampling distances according to equation (29), we again generate snapshots of the hole configurations. Note that, although in the mean-field theory fluctuating stripes pinned to the centre were described, the classical formulation as given above captures stripes that are not pinned to the boundary and hence naturally form extended hole configurations (see also Fig. 4a). We compare the results with the experimental data in Extended Data Fig. 9g, in which, for $k_BT/J_y = 0.8$, we observe similar features to the experiment and the results from mean-field theory for $k_BT/t_x = 0.36$. Finally, we investigated the mean length of the excess stripes in the MHZ approach as a function of temperature (see Extended Data Fig. 9e). We observe an increase of this length below $J$, marking the onset of stripe-like structures.

**Limitations of the effective descriptions.** Both descriptions of the fluctuating stripe presented above are approximate, as they rely on assuming an effective confining potential (equation (21)) between

holes in neighbouring chains. The latter description is derived within the geometric string approach, that is, assuming that the fluctuating charges merely displace spins in the background when they move around, without affecting their spin correlations. At low temperatures, this is a valid assumption, whereas at higher temperatures—when longer strings play an increasingly important role—we expect corrections to the confining potential. In particular, the question remains whether an unbinding transition of holes out of the stripe can take place. Latest for $T \to \infty$ in the effective model this is expected to happen, for which spin correlations in the background $C_d \to 0$ and thus $V_{pot}(\Sigma) \equiv 0$ in equation (21).

Another limitation of the effective models of an individual stripe is its limitation to one hole per chain. On one hand, extended interactions between neighbouring stripes at low temperatures can lead to ordering and the formation of a stripe phase with long-range charge and (incommensurate) spin order. On the other hand, at higher temperatures, the spatially extended nature of the individual stripes can cause interaction effects between them to play a role in the expected thermal unbinding transitions into a deconfined chargon gas[65].

## Data availability

The datasets generated and analysed during the present study, as well as the code used for the analysis, are available from the corresponding author on reasonable request. Source data are provided with this paper.

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

**Acknowledgements** We thank E. Demler for fruitful discussions. This work was supported by the Max Planck Society (MPG), the Horizon Europe programme HORIZON-CL4-2022 QUANTUM-02-SGA (project 101113690, PASQuanS2.1), the German Federal Ministry of Education and Research (BMBF grant agreement 13N15890, FermiQP) and Germany's Excellence Strategy (EXC-2111-390814868). T.C. acknowledges support from the Alexander von Humboldt Foundation. F.G. acknowledges support from the European Research Council (ERC) under the European Union's Horizon 2020 research and innovation programme (grant agreement no. 948141) from ERC Starting Grant SimUcQuam.

**Author contributions** D.B. led the project. T.C. and D.B. contributed substantially to data collection and analysis. H.S., A.B. and F.G. performed the theory calculations. D.B. and T.A.H. wrote the manuscript. T.A.H. and I.B. supervised the study. All authors worked on the interpretation of the data and contributed to the final manuscript.

**Funding** Open access funding provided by Max Planck Society.

**Competing interests** The authors declare no competing interests.

**Additional information**
**Correspondence and requests for materials** should be addressed to Dominik Bourgund or Timon A. Hilker.

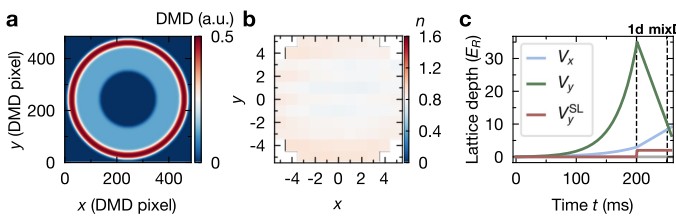

**Extended Data Fig. 1 | Lattice potential and ramps. a**, Pattern applied to the digital mirror device for potential shaping. **b**, Resulting density profile in the centre. **c**, Lattice ramps to prepare the mixD system. We first ramp in 200 ms to decoupled 1d chains before ramping to the full mixD system.

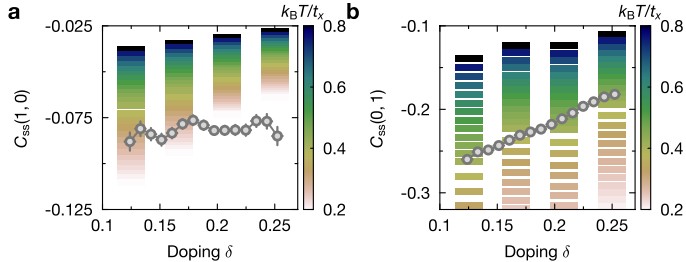

**Extended Data Fig. 2 | Spin correlations as a function of doping.** Nearest-neighbour spin correlations along $x$ (**a**) and $y$ (**b**) for different doping levels. We compare the experimental data (grey markers) to numerical data for $C_{ss}(1, 0)$ for different temperatures for simulations on $L_x, L_y = 8, 3$ and $J_y/t_x = 0.5$ to get an estimate for our temperature.

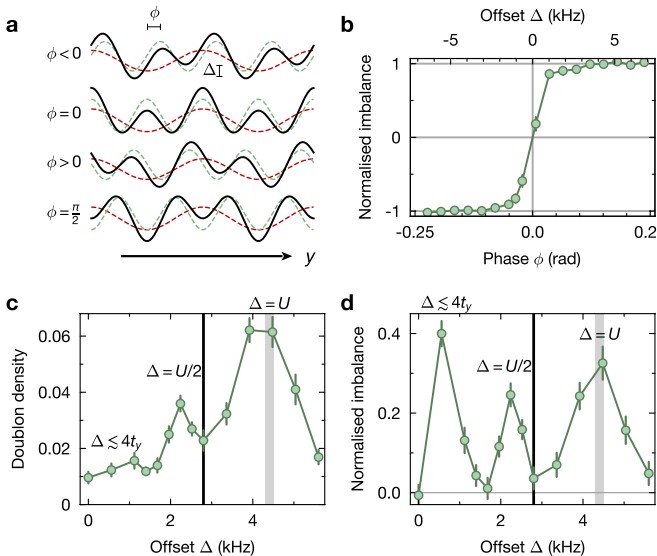

**Extended Data Fig. 3 | Offset calibration. a**, We load a dilute cloud into a deep superlattice ($V_x = 40E_R$, $V_y = 8E_R$ and $V_y^{SL} = 21E_R$) with different phase and extract the imbalance in occupation between neighbouring chains to calibrate the symmetric phase. **b**, At the symmetric double-well configuration ($\phi = 0$), we reach zero imbalance, whereas even for small deviations, we quickly occupy only one part of the double well. All of the main experimental results are obtained for $\phi = \pi/2$. Doublon density (**c**) and imbalance between chains (**d**) as a function of potential offset $\Delta$ (that is, superlattice power) for a relative superlattice phase of $\pi/2$. The peak in the doublon density coincides with the interaction energy $U$ (grey line), at which atoms are then resonantly transferred to neighbouring chains. For small offsets, tunnelling is not yet fully suppressed and an imbalance is created. Above an intermediate peak at $U/2$ (created by a higher-order process), there is a low-imbalance regime in which the experiment is performed (black line).

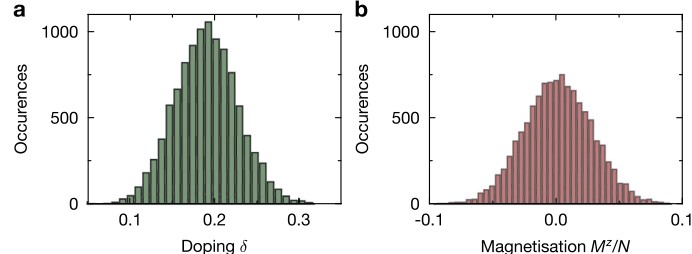

**Extended Data Fig. 4 | Data statistics.** Histograms of doping (**a**) and magnetization (**b**). We take data between 10% and 30% doping, whereas the total magnetization is well centred around 0.

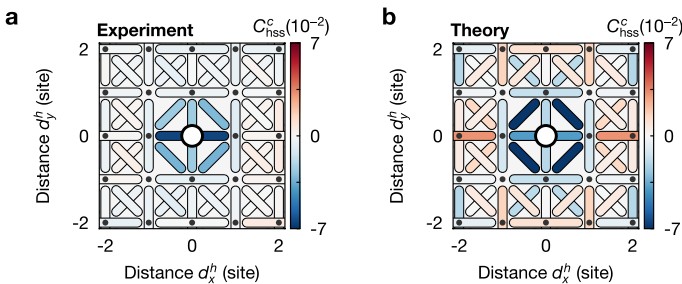

**Extended Data Fig. 5 | Connected three-point correlator. a**, Fully connected, symmetrized, three-point hole–spin–spin correlator. By removing the AFM background, we focus on the extra effect introduced by the dopant that is compatible with the onset of a domain wall in the local AFM pattern across the dopant. **b**, A comparison with DMRG calculations at $k_B T/t_x = 0.4$ shows qualitatively similar results.

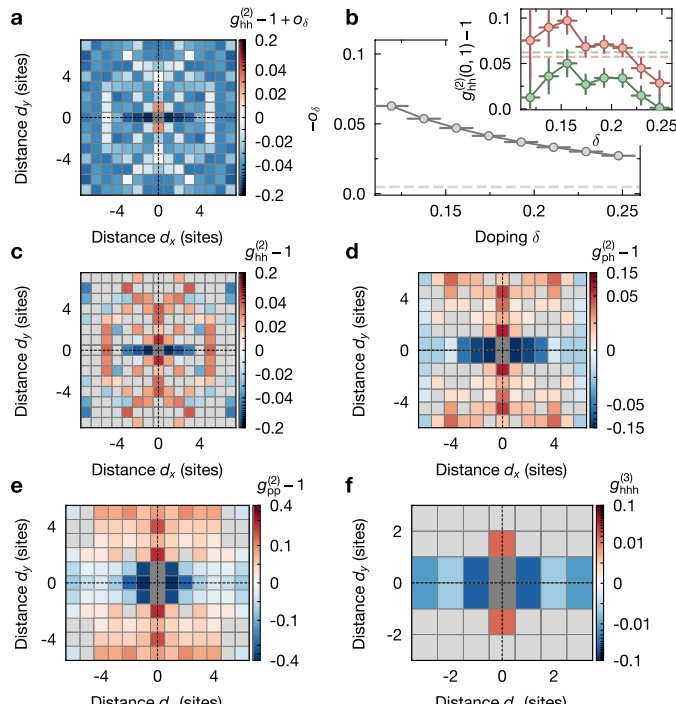

**Extended Data Fig. 6 | Correlator offsets and significance. a**, Correlation map of Fig. 2a without offset correction. **b**, Correlator offset $o_\delta$ as a function of doping. The nearest-neighbour hole–hole correlator as a function of doping with (red) and without (green) offset correction is shown in the inset. The horizontal dashed lines are the same correlator without binning by density, in which case the offset almost vanishes (dashed line in **b**). Symmetrized hole–hole (**c**), pair–hole (**d**), pair–pair (**e**) and hole–hole–hole (**f**) correlation maps with errors. All values consistent with zero are set to grey. The signals discussed in the main text are all still clearly visible.

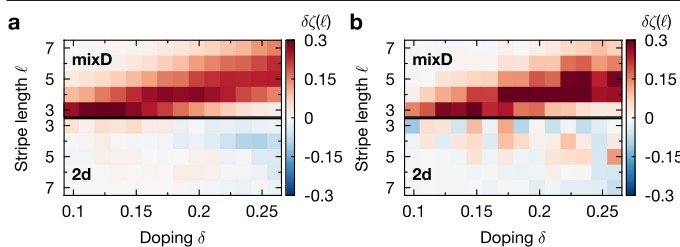

**Extended Data Fig. 7 | Stripe-length random data comparison. a**, Comparing experimental data with randomly generated data without any correlations. **b**, Comparison with randomly placed pairs within the system. Both methods yield qualitatively the same result as the data in the main text.

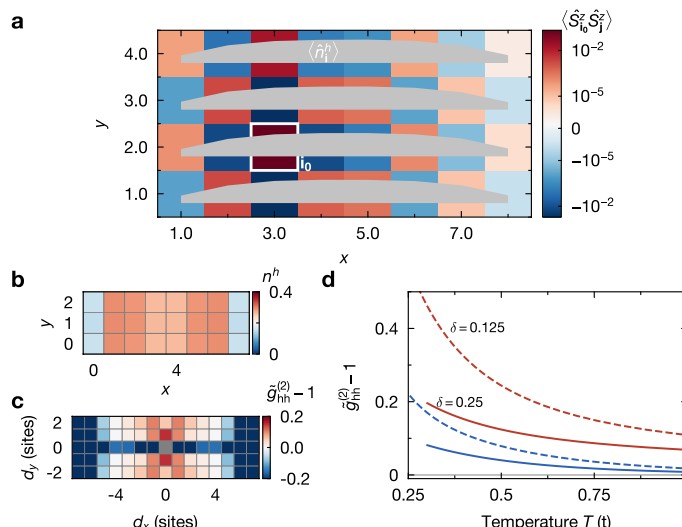

**Extended Data Fig. 8 | Finite-temperature DMRG. a**, DMRG calculations for a $L_x \times L_y = 8 \times 4$ system with periodic boundaries along the short direction, temperature $k_B T/t_x \approx 0.4$ and Hamiltonian parameters as in the experimental setup. Shown are the on-site hole density distributions in each leg, $\langle \hat{n}_i^h \rangle$ (grey lines), as well as spin–spin correlations $\langle \hat{S}_{i_0}^z \hat{S}_j^z \rangle$ (colour-coded) for reference site $\mathbf{i}_0 = [x=3, y=2]$ (white box). At the maximum hole density distribution in the centre of the chain, a domain wall of the AFM background forms, that is, a single stripe is observed. **b**, Hole density for $L_y = 3$, $\delta = 0.25$, $k_B T/t_x = 0.41$. Two separate stripes form at this doping level. **c**, Hole–hole correlations versus distance, reminiscent of the structure shown in Fig. 2a. **d**, Hole correlations as a function of temperature for $d_y = 1$ (red) and $d_y = 2$ (blue), $\delta = 0.125$ (dashed lines) and $\delta = 0.25$ (solid lines).

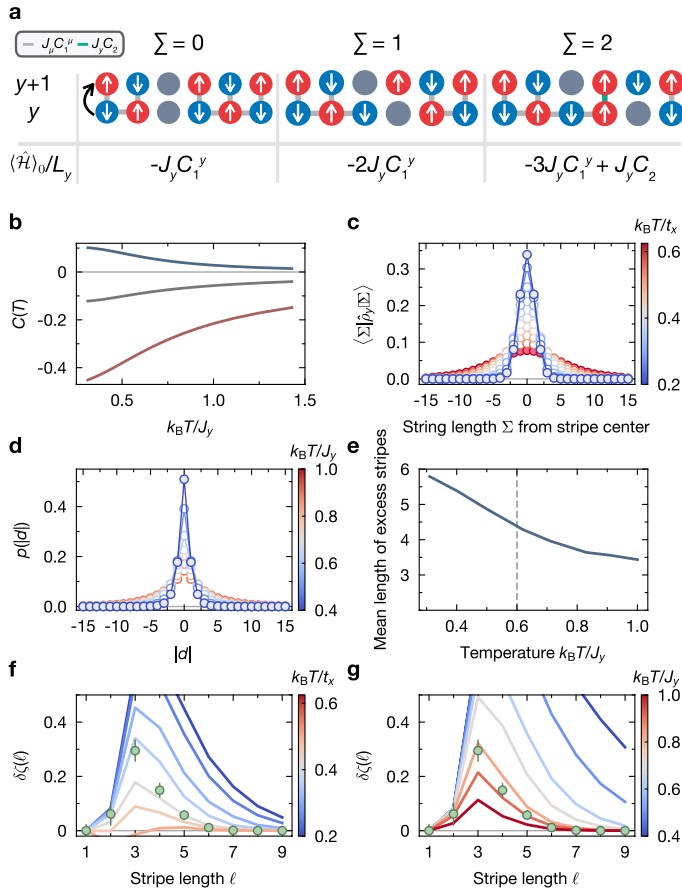

**Extended Data Fig. 9 | Effective models. a**, Illustration of the effective potential between chain $y$ with its neighbouring chain $y + 1$. Grey lines illustrate energy contributions proportional to $J_\mu C_1^\mu$, $\mu = x, y$; green line denotes diagonal correlation with energy contributions of order $J_y C_2$ starting at $|\Sigma| \geq 2$. Intrachain energy corrections from the Néel state of strength $J_x C_1^x$ are constant and not written down explicitly in the potential. **b**, Thermally averaged two-point correlations of the undoped Heisenberg model $C_1^\mu(T) = \langle \hat{\mathbf{S}}_i \cdot \hat{\mathbf{S}}_{i+\mathbf{e}_\mu} \rangle_T$, $\mu = x$ (grey), $y$ (red) and $C_2(T) = \langle \hat{\mathbf{S}}_i \cdot \hat{\mathbf{S}}_{i+\mathbf{e}_x+\mathbf{e}_y} \rangle_T$ (blue) calculated from DMRG calculations for $J_x/J_y = 0.3$ on a $12 \times 4$ lattice with periodic boundary conditions applied along the short ($y$) direction. **c**, Thermally averaged string-length distribution $\langle \Sigma | \hat{\rho}_{MF}^{(0)} | \Sigma \rangle$ for temperatures $k_B T/t_x = [0.2, 0.625]$ and $t_x/J_y = 2$ using the thermal correlations in the Heisenberg model in **b**. **d**, Hole distance distributions in the MHZ approach (equation (29)) for various temperatures $k_B T/J_y = 0.4$–0.9. **e**, Mean length of excess stripes as calculated from the MHZ approach as a function of temperature. The dashed line marks the experimental temperature. **f**, Difference in stripe lengths from mean-field theory to random distribution for temperatures $k_B T/t_x \in [0.2, 0.625]$ and $J_x/J_y = 0.3$ and experimental data for $\delta = 0.111$ (markers) as in the inset of Fig. 4a. **g**, Stripe-length histograms using the classical MHZ estimate for temperatures $k_B T/J_y \in [0.4, 1]$, which shows qualitatively similar results.