## [Peer Review File · Nature]

Manuscript Title: Formation of individual stripes in a mixed-dimensional cold-atom Fermi-Hubbard system

Reviewer Comments & Author Rebuttals

Reviewer Reports on the Initial Version:

Referee #1 (Remarks to the Author):

The article reports the experimental results in a mixed-dimensional Fermionic Hubbard model on optical lattice. The experiment is clearly an extension of the previous paper of the same group, Nature 463–467 (2023) on ladder configuration to two dimensional configuration. The major observation is the formation of stripe of holes along the y direction.

The charge stripe has indeed been widely discussed in the high temperature superconductor context. To my best knowledge there is no observation of stripe in cold atom optical lattice. Therefore the first clear evidence of stripe order in optical lattice is an important progress.

However, I am not convinced that the current experiment really has solid evidence of a stripe order for the following reasons:

(1) Stripe order is most precisely defined as a peak $q=(Q,0)$ for the Fourier transformation of the density density correlation function $\langle n_h(x,y)n_h(0,0) \rangle$. The basic picture is that the holes concentrate along a stripe in y direction and the stripes are organized in a periodic way in x direction with wavelength $2\pi/Q$.

The current article does not have any indication of any periodic structure with momentum Q along x direction at all. So this does not seem to be the stripe order discussed in the high T_c superconductor context.

(2) The major evidence is that the density-density correlation along y direction is positive. In the mixed dimensional setup, because the hopping t_y is turned off, indeed the holes are favored to stay together along y direction. This result was already demonstrated in the previous paper Nature 463–467 (2023). It is good that this 2D experiment confirms that the behavior extends to 2D, but I'm not sure the evidence is enough to prove the existence of a stripe.

(3) The most direct, unambiguous evidence of stripe formation is to just provide counts of a stripe of holes in the snapshots. The authors report this in Fig.4 a . From my eyes, the number of 'stripes' is not significantly larger than that from the usual 2D model and random distribution.

There is of course some excess because of the positive density correlations along y direction. But the effect is not large enough to claim the formation of well-defined stripe. The probability of finding a stripe with length larger than 5 is almost zero.

In summary, I think the current data is not enough to claim the existence of well-defined stripe order. The results mainly confirm the positive density correlation along the y direction, extending the same conclusion of the previous paper Nature 463–467 (2023). I do not recommend the current manuscript to be published in Nature.

Referee #2 (Remarks to the Author):

In this work, the authors study the formation of extended density structures in a doped mixed-dimensional system realized with ultracold fermions in an optical lattice. This mixed-dimensional system consists of one-dimensional chains subject to a staggered potential which suppresses tunnelling between the chains while preserving magnetic coupling, thereby effectively mediating an effective attraction between holes across the chains. While hole-hole correlations display antibunching along the chains due to Pauli repulsion, they show bunching beyond nearest-neighbour sites in the transverse direction, leading to structures that the authors identify as stripes. Long-range hole attraction is confirmed in measurements of pair-hole and pair-pair correlations (Fig. 3) and histograms of the stripe length (Fig. 4). Spin correlations across the holes are furthermore reminiscent of a change of AFM domain phase (Fig. 5).

Compared to their previous work [45], which demonstrated hole pairing in a two-leg ladder, the authors have implemented a cold-atom mixed-dimensional model over impressive system sizes totalling more than 100 sites. This achievement reduces finite-size effects which would otherwise prevent the observation of long-range hole attraction, and allows to truly enter a regime where quantum simulation has an edge over ab-initio numerical simulations. However, I find the presentation of the manuscript misleading by drawing a strong connection between the mixD system studied here and stripe order in cuprate materials. Mixed-dimensional systems are anisotropic systems that are finely tuned to favour density ordering along one direction at the expense of magnetic ordering along the other, at entropies currently achieved in cold-atom systems ([45] is referred to as a "tailored ladder system"). As such, the anisotropic t-J model studied here differs from the (isotropic) lattice models that have been the focus of extensive work in connection to cuprate materials [1-12] and where the origin of stripes is unclear. Contrary to what is hinted at the beginning of the abstract, the current manuscript does not address the central question of how mixD stripes are adiabatically connected to stripes encountered in isotropic models, and how they compete with other ordered phases.

That being said, I find the connection between the current cold-atom mixD system and nickelate materials intriguing (to which the authors state that "More generally, [their] approach has direct relevance" L.37-38). My concerns about the presentation would be alleviated if the focus of the abstract (essentially the first four sentences) and of the first introduction paragraph were shifted away from cuprates to materials where "mixed dimensions play an essential role". In addition, the mixed-dimensional nature of the current experimental setup should also be emphasized in the "Here we" abstract sentence and the discussion of the state-of-the-art in the second introduction paragraph. I still expect that motivating the experimental results in the context of mixD materials would still be of utmost interest for the broad readership of Nature, given their relevance to very recent condensed-matter experiments on nickelates [14-17]. It would further raise interest from the AMO quantum simulation community about these exotic superconductors.

Besides this point, I recommend the authors address the following questions before further consideration:

1. The abstract mentions that the experiment "access[es] the interesting crossover temperature where stripes begin to form [13]", but this crossover versus temperature is not discussed later in the text. Could the authors elaborate on why this temperature regime is interesting? What are the relevant temperature scales for pair resp. stripe formation? It would be good to explicitly mention these energy scales when introducing the t-J model and experimental temperatures in paragraph L.123-138. Although experimental data is obtained at fixed temperature, a qualitative discussion of two-point correlators and string-length distribution versus temperature could be added based on numerical data shown in Figs. S13-16.

2. What is the reason for showing hole-hole correlations in Fig. 2 at fixed doping $\delta = 0.18$? Post-selection on atom number seems to introduce a global offset comparable to the positive correlators reported at $d_x = 0$, and hole attraction would not be observed beyond nearest neighbours without compensating for this offset (Fig. S7). How does the hole-hole correlation map look like without post-selection (such as in Fig. 3)?

3. The correlation maps display additional features that are not discussed in the text, such as positive Y-shaped features in Fig. 2a above and below $d = (0, 0)$, and positive pair correlators at $|d_y| = 3, 4$ in Figs. 3b and c. Are these features a physical indication of fluctuations along x of the stripe position, or of edge effects? Or do they result from experimental limitations, such as potential inhomogeneities? These features are at least as statistically significant as the positive signal pointed out at $d_x = \pm 5$ and are therefore worth commenting on.

4. On a related note, it would be important to provide quantitative estimates of the system homogeneity, and to state whether this affects the observed density correlations. For instance, a residual harmonic confinement or anti-confinement along x, together with total atom number fluctuations, could also explain long-range connected density correlations along the y direction. In addition, is the distribution of the stripes detected in Fig. 4 uniform along the x direction?

5. Additional comments are needed to clarify the claims of the existence of a second stripe or a charge density wave (Lines 164-167 and 228-232):

a. Fig. 2: Is the positive signal at $d_x = \pm 5$ visible at dopings other than $\delta = 0.18$? If it is, is its position consistent with the expected period of the charge density wave $1/\delta$?

b. Fig. 2: Shouldn't the positive hole-hole correlations at $d_x = \pm 5$ entail the existence of negative correlations at $d_y \neq 0$ and $1 \leq |d_x| \leq 4$ due to Pauli repulsion between the two stripes?

c. Figs. 3b and c: I do not really understand why the average of the correlators along y is interpreted as the existence of a charge structure at $d_x = 4$. It seems here to result from a fortuitous combination of strong Pauli repulsion on the central lines $|d_y| \leq 1$, which dominates the average at $|d_x| \leq 3$, and a broad positive background on the outer lines $|d_y| > 1$, which

dominates the average at $|d_x| \geq 4$. Wouldn't a certain type of long-range anisotropic correlations without stripe order lead to a similar average? There is no proper density wave structure visible for any individual line, what about after binning with respect to doping like in Fig. 2?

6. Unless there is strong evidence for a second stripe, the plural word "stripes" in the title is potentially misleading, as it could be understood as the observation of several stripes in individual experimental realizations, instead of the observation of a single stripe in correlations measured over repeated experimental realizations.

7. As far as I understand, the spin-correlation function between next-nearest neighbours along x is compatible with what is expected for decoupled 1d chains with antiferromagnetic correlations (Ref. [51]). In contrast, negative spin correlations on diagonal neighbours around the hole are reminiscent of the spin texture of a magnetic polaron also observed in the same group (Ref. [38]). Are there indications of longer-range domains along the stripe direction in the hole-spin-spin correlations? For example, does the presence of a hole at $d = (0, 0)$ increase AFM correlations along vertical nearest-neighbour bonds $(1, j) \leftrightarrow (1, j+1)$?

Minor comments:

8. L.144, "Due to the finite size and particle number fluctuations": can the authors give more intuition on the systematic offset? In the absence of fluctuations, where exactly N atoms are independently and identically distributed over system size V , the conditional probability of finding a particle at any site j given a particle is located at site $i \neq j$ is reduced by $1/V$ (since $N-1$ particles, not N , are distributed over all sites other than i), which would lead to a negative g_2 -Function equal to $-1/N$? Conversely, one would expect that large particle number fluctuations to lead to positive long-range density-density correlations. These two limits could be more explicitly explained in Methods D2.

9. L.146: since the offset δ_0 is doping-dependent, why not indicating a range $\delta_0 \sim -0.03$ to -0.06 (according to Fig. S7b)? What is the experimental error on this offset?

10. L.280-282, "While our parameter regime is not favourable to investigate structure factors": could the authors clarify this statement?

11. L.297-299 and L.325-329: is there any theory reference describing the "change in the parity of the AFM pattern across impurities"? Only Ref. [51] from the same group is cited.

12. L.335-339, "we observe these features without long-range AFM correlations": could the authors indicate spin-spin correlation lengths along the x - and y -directions?

Referee #3 (Remarks to the Author):

The manuscript "Formation of stripes in a mixed-dimensional cold-atom Fermi-Hubbard system" investigates the properties of doped correlated systems from cold-atom experiments, with the main goal of examining the formation of stripes. As the energy scale to access stripes is usually very low, so far inaccessible for the current 2D implementations, the authors propose a mixed-dimensional (mixD) setting. In words, such a mixD system is composed of weakly coupled chains, where interchain hoppings are very unlikely. On the other hand, the interchain exchange is maintained in both directions. Within this setup, the authors examine two and three-particle correlation functions, with their main results summarized in Figs. 2-5, and provide direct evidence of the formation of stripes from a strong spin-charge correlation among chains. To support their experimental findings, the authors also performed finite temperature density matrix renormalization group simulations for the t-J model.

The results are solid and sound, and they further clarify the role of spin-charge correlations in the formation of stripes. Furthermore, the comparison between the mixD system with the isotropic 2D and random distribution holes case shows a clear evidence of stripe formation. Despite this, there are some points that deserve clarification, which I list below.

1. It is not clear why the authors presented the DMRG for the largest lattices (8x4) in the SI, while in the main text they show results to smaller ones (8x3).
2. When discussing Figure 2.d, the authors mention that the theoretical results agree "qualitatively" with the experimental ones. This claim is incompatible with Fig.2. For instance, the (0,2) case is negative for the experimental data, but positive for the DMRG ones. This is a clear qualitative and quantitative difference. The authors explain this inconsistency by mentioning that it may be due to finite-size effects. If so, I return to item 1: why was the largest lattice in the SI and not in the main text?
3. Suppose we agree that strong finite-size effects are leading to this quantitative disagreement on theoretical and experimental correlation functions (even for nearest neighbors), as shown in Fig.2, and as emphasized by the authors. Then, it leads me to another question: why should one believe that the temperature estimation by DMRG, provided in Fig.S2 (probably for an 8x3 lattice), has not suffered from strong finite-size effects? From the theoretical-experimental inconsistencies presented in Fig.2, my partial conclusion is that one has many reasons to believe that the temperature is not well defined in the paper, or at least its error bars should be much larger than those presented.
4. If the temperature estimation of DMRG is correct, it would be worth it if the authors could provide an estimation for the temperature at which the stripes appear in the 2D case. With this, the community would know clearly how challenging this task is.

Minor points:

5. The authors mention in the text that the engineering of the mixD setting just "slightly" increases the spin coupling in the perpendicular direction. However, from the data reported in the SI the one has $J_y/J_x \approx 4$. It may be a slight increase in J_y , but the final exchange anisotropy is large. I recommend removing the "slightly" in the phrase, to avoid misleading conclusions.

6. The acronym DRMG is not defined in the main text.

Author Rebuttals to Initial Comments:

Introductory statement

The authors thank the referees for their time and detailed feedback, which has been very helpful in further improving the manuscript. Their comments allowed us to more precisely differentiate between the ordered stripe phase found in solid-state materials and the individual stripes of holes that form in our experiment. Furthermore, we are grateful for the advice to highlight more prominently the connection of our results to novel superconducting materials (nickelates).

We, furthermore, would like to stress that our experiment explores an intermediate temperature regime, which is only poorly understood both in experiments and in numerics: At high temperatures (much higher than J), the dopants are effectively free particles. At slightly lower temperatures around J , magnetic polarons form, which have a higher mass but are still independent. Here, at $T \approx J/2$, we explore the regime in which magnetically mediated interactions between polarons start to dominate (in mixed-dimensional systems). However, even lower temperatures, likely below $0.1J$, are required for the fully ordered stripe phase in plain 2d Fermi-Hubbard systems. The latter can be experimentally studied in condensed matter systems via the structure factor in both the charge and spin sector. The onset of extended charge structures, on the other hand, requires microscopic correlation measurements, which we present in the manuscript. We call these structures "individual stripes" and have carefully revised the manuscript to avoid confusion with the ordered striped phase, which we do not claim to observe.

Our geometry (mixed-dimensional with 2D-spin and 1D-charge couplings) is located between certain cuprates (believed to be described by the plain 2d Hubbard model) and nickelates (mixed-dimensional with bilayer-spin and 2d-charge). These regimes are adiabatically connected but show different physics e.g. strong singlets in the bilayer (and ladder) systems but medium-range AFM correlations in the 2D systems. These connections can be explored in detail with future quantum simulation experiments, especially at slightly lower temperatures where we can study signs of long-range charge order and the thermodynamics of fluctuating stripes. Here we demonstrate that such mixed-dimensional experiments are possible in a regime that is intractable for numerics, and we observe for the first time extended charge structures due to magnetically mediated attractions.

Changes to the Manuscript

- Adapted abstract and introduction to focus more on mixed-dimensional materials like nickelates
- Improved introduction to distinguish more clearly between mixed-dimensional and normal 2d systems
- Improved distinction between the observed individual stripes and the stripe phase expected at lower temperatures
- Explicit mentioning of temperature where stripes are expected to appear from numerics
- Improved focus on relevance of experimental temperature regime
- Added range of offset values to main text
- Improved interpretation of signals pointing towards multiple stripes
- Added definition of DMRG
- Improved Fig. 5b and the definition of its correlator
- Corrected doping level stated in description of Fig. 4a

Changes to the Methods

- Added comment on spin correlation length
- Changed figure S1 to add experimental density
- Added comment on the effect of numerical system size for temperature estimation
- Added temperature dependence of mean length of excess stripes to figure S16
- Added details on limits of effective numeric descriptions

Response to report from the first referee

We are grateful for the referee's insights into the unambiguous identification of stripe phases. We improved the manuscript according to their suggestions by clarifying the connection between our results and stripe phases in solid-state materials.

The article reports the experimental results in a mixed-dimensional Fermionic Hubbard model on optical lattice. The experiment is clearly an extension of the previous paper of the same group, Nature 463–467 (2023) on ladder configuration to two dimensional configuration. The major observation is the formation of stripe of holes along the y direction.

The charge stripe has indeed been widely discussed in the high temperature superconductor context. To my best knowledge there is no observation of stripe in cold atom optical lattice. Therefore the first clear evidence of stripe order in optical lattice is an important progress.

However, I am not convinced that the current experiment really has solid evidence of a stripe order for the following reasons:

1. Stripe order is most precisely defined as a peak $q = (Q, 0)$ for the Fourier transformation of the density density correlation function $\langle n_h(x, y)n_h(0, 0) \rangle$. The basic picture is that the holes concentrate along a stripe in y direction and the stripes are organized in a periodic way in x direction with wavelength $2\pi/Q$. The current article does not have any indication of any periodic structure with momentum Q along x direction at all. So this does not seem to be the stripe order discussed in the high T_c superconductor context.

We agree with the referee's assessment that stripe *phases* are commonly identified via charge and spin structure factors. Therefore, as the referee states, we cannot claim to observe a full stripe phase. Instead, in our work, we focus on the formation of individual stripes from a sea of holes, which we see as an important and necessary precursor to the formation of the ordered phase. In our parameter and temperature regime, these stripes are mostly independent of each other.

Through our observations, we conclude that we created a state in which extended structures of holes form. These structures, which repel each other but do not order yet (due to system size and temperature), we call (individual) stripes because of their fluctuating one-dimensional form. In that sense, it may be interpreted as an analogue to a state of preformed pairs above T_c of a superconductor. We furthermore note that our spin-couplings are very anisotropic, which further reduces the ordering along x that would be measured in the structure factors.

Therefore, following the argument of the referee, we improved the language of our manuscript. We now focus on calling our charge structures "individual stripes" and explicitly state that this does not refer to an ordered stripe phase.

2. The major evidence is that the density-density correlation along y direction is positive. In the mixed dimensional setup, because the hopping t_y is turned off, indeed the holes are favored to stay together along y direction. This result was already demonstrated in the previous paper Nature 463–467 (2023). It is good that this 2D experiment confirms that the behavior extends to 2D, but I'm not sure the evidence is enough to prove the existence of a stripe.

The referee correctly stated that this work extends some of the elements of our previous work on mixed-dimensional 2-leg ladder systems. As the referee notes, we are able to go beyond those results and experimentally confirm that larger, two-dimensional systems with suppressed hopping along one direction still exhibit effective attraction of charge carriers. Furthermore, charge pairs are not repelled from each other, but instead form larger structures perpendicular to the direction of charge motion. This is supported by measurements of 3- and 4-point charge correlations and histograms of stripe-like structures.

Also, from a theoretical point of view, ladders and extended 2d Hubbard systems are very different. In ladders, strongly bound singlets form in the spin sector, leading to a spin gap. The breaking of these pairs is the dominant process that ultimately favors bound pairs of holes. This is well captured in numerical simulations, for example MPS calculations converge for relatively small bond dimensions. However, in the 2d plane, the spin correlations exhibit a more long-ranged character, and there is no spin gap at low doping. Even in small systems (8x3), the numerical simulation with matrix product states requires huge bond dimensions of 30000, placing the experimental system size far out of reach. Therefore, in contrast to ladder systems, the formation of individual stripes in 2d systems with mixed-dimensional couplings is not obvious and goes significantly beyond previously published results by our team.

Mixed-dimensional measurements may lead to insights into the plain 2d FHM due to a possible direct connection between the two: by adiabatically reintroducing tunnelling along the y -direction, the ground state of the 2d system can potentially be reached from an extended mixed-dimensional system, and the stripes observed in our present manuscript may be adiabatically connected to the stripe phase of the plain Hubbard model.

Finally, we emphasise that our implementation is directly related to recently discovered high-temperature superconducting nickelates [1], which feature a mixed-dimensional bilayer system and establish mixed-dimensional systems as a promising new route to realise high- T_c materials. Therefore, by showing the existence of individual stripes, we establish quantum simulation as a tool to explore extended fluctuating charge structures.

3. The most direct, unambiguous evidence of stripe formation is to just provide counts of a stripe of holes in the snapshots. The authors report this in Fig.4 a . From my eyes, the number of 'stripes' is not significantly larger than that from the usual 2D model and random distribution. There is of course some excess because of the positive density correlations along y direction. But the effect is not large enough to claim the formation of well-defined stripe. The probability of finding a stripe with length larger than 5 is almost zero.

We thank the referee for their comment on our stripe-counting results. Following the response to the first point, it becomes clear that we refrain from identifying these structures as a striped phase. Instead, we count the structures of multiple holes and study their shape. The results are consistent with the expected formation of individual stripes extending along y -direction. Although the absolute values of the signal may seem relatively small, they are very significant when compared to the hole correlator values of Fig.2. By comparing the measurement with different random distributions, including a random distribution of pairs in Fig. S9 (which have much higher correlations than in the experiment), the clear excess of stripe-like structures is shown.

Even in the ground state of a perfect single stripe, as described by our mean-field ansatz, the probability of observing very long stripe-like structures (as defined in our manuscript) is small due to quantum fluctuations associated with the hole motion along the x -direction. The details of the counting depend on the definition of a continuous stripe as extracted from snapshots. Here we resorted to a simple, experimentally motivated definition based on no more than one fluctuation along x .

Finally, we note that the probability of forming large stripes is largely doping-dependent, which is why we compare our results to randomly distributed holes. Longer stripe-like structures emerge in our system for larger and larger doping levels.

Response to report from the second referee

We are thankful for the referee's insightful comments, which greatly improved the quality of the manuscript. We appreciate their appreciation of our work's relevance to novel high-temperature superconducting materials. Following their comments, we adapted the manuscript to focus further on the mixed-dimensional character and its relevance to solid-state materials, and we clarified the interpretation of our correlation data. Here, we answer their comments in detail.

Compared to their previous work [45], which demonstrated hole pairing in a two-leg ladder, the authors have implemented a cold-atom mixed-dimensional model over impressive system sizes totalling more than 100 sites. This achievement reduces finite-size effects which would otherwise prevent the observation of long-range hole attraction, and allows to truly enter a regime where quantum simulation has an edge over ab-initio numerical simulations. However, I find the presentation of the manuscript misleading by drawing a strong connection between the mixD system studied here and stripe order in cuprate materials. Mixed-dimensional systems are anisotropic systems that are finely tuned to favour density ordering along one direction at the expense of magnetic ordering along the other, at entropies currently achieved in cold-atom systems ([45] is referred to as a "tailored ladder system"). As such, the anisotropic t-J model studied here differs from the (isotropic) lattice models that have been the focus of extensive work in connection to cuprate materials [1-12] and where the origin of stripes is unclear. Contrary to what is hinted at the beginning of the abstract, the current manuscript does not address the central question of how mixD stripes are adiabatically connected to stripes encountered in isotropic models, and how they compete with other ordered phases.

That being said, I find the connection between the current cold-atom mixD system and nickelate materials intriguing (to which the authors state that "More generally, [their] approach has direct relevance" L.37-38). My concerns about the presentation would be alleviated if the focus of the abstract (essentially the first four sentences) and of the first introduction paragraph were shifted away from cuprates to materials where "mixed dimensions play an essential role". In addition, the mixed-dimensional nature of the current experimental setup should also be emphasized in the "Here we" abstract sentence and the discussion of the state-of-the-art in the second introduction paragraph. I still expect that motivating the experimental results in the context of mixD materials would still be of utmost interest for the broad readership of Nature, given their relevance to very recent condensed-matter experiments on nickelates [14-17]. It would further raise interest from the AMO quantum simulation community about these exotic superconductors.

We agree with the referee's observation that the mixed-dimensional systems investigated here are more closely related to the novel class of nickelate material rather than the cuprates, due to the mixed-dimensional couplings. On the other hand, the undoped parent spin state considered in our work features two-dimensional antiferromagnetism rather than rung-singlet formation, bringing it closer to the physics of cuprates. As such, our manuscript addresses a regime between the bilayer nickelates (believed to favor pairing) and the single-layer cuprates (believed to favor stripes). In the future we propose to experimentally tune tunnel couplings in such a way that would allow to adiabatically deform our present system into either of the two settings resembling bilayer nickelates or cuprates.

We thank the referee for sharing their enthusiasm for the present work to be 'of utmost interest for the broad readership of Nature'. Following the suggestion of the referee, we adapted our

abstract and introduction to clarify this issue while emphasising the connection of our work to nickelate compounds. We thereby stressed the mixed-dimensional nature of our work and additionally highlighted the connection between plain 2d and mixD systems in the conclusion.

1. The abstract mentions that the experiment "access[es] the interesting crossover temperature where stripes begin to form [13]", but this crossover versus temperature is not discussed later in the text. Could the authors elaborate on why this temperature regime is interesting? What are the relevant temperature scales for pair resp. stripe formation? It would be good to explicitly mention these energy scales when introducing the t-J model and experimental temperatures in paragraph L.123-138. Although experimental data is obtained at fixed temperature, a qualitative discussion of two-point correlators and string-length distribution versus temperature could be added based on numerical data shown in Figs. S13-16.

We thank the referee for identifying this missing discussion. For our current temperature and parameter regime, we first observe evidence of individual charge-ordered structures forming, which may be seen as a precursor to a fully ordered *stripe phase*. The question of how to best characterise this regime – where thermal fluctuations compete with the tendency to form first individual stripes and then, at lower temperatures, lock these individual stripes into place along x – has so far not been studied in detail on a microscopic level as it presents a significant computational challenge. While the cited theoretical studies investigated emerging signals in charge and spin structure factors, the formation of *individual* stripes that do not yet form a full charge density wave is an intriguing effect that we studied here in systems extended along both spatial directions.

Following this, we added further details to the manuscript on the relevant energy scales, where stripe order in plain 2d FH systems is expected based on past numerical studies at temperatures below $0.05 t$. When using mixed-dimensional systems, this energy scale increases by a factor of approximately 5-10 [2] (where quantitative comparisons are difficult due to the very limited numerical system size that can be handled), underlining the power of the mixD approach. We also refer to the introductory statement for further elaboration on the relevant energy scales.

The numerical data shown in S13-16 is based on an effective model. This model was exceptionally useful in our case, as it provided the only point of comparison to our stripe length analysis, which required larger system sizes than are available for MPS-simulations. However, it may not be able to provide further insight into the physics of the full mixD $t - J$ model. We therefore restrict the interpretation of this effective model to regimes where DMRG is no longer applicable.

We conducted further investigations into the temperature dependence based on the classical MHZ model by calculating the mean length of the excess stripes as a function of temperature, i.e. taking the mean of the data presented in Fig. S15b (shown here in Fig. R1). We observe a significant increase below J , marking the onset of stripe-like structures. We have added the figure to the Supplementary Materials (Fig. S16c).

2. What is the reason for showing hole-hole correlations in Fig. 2 at fixed doping $\delta = 0.18$? Post-selection on atom number seems to introduce a global offset comparable to the positive correlators reported at $d_x = 0$, and hole attraction would not be observed beyond nearest

Figure R1: **Mean length of excess stripes from effective models.** Dependence of mean length of excess stripes as calculated from the classical MHZ approach as a function of temperature. The dashed lines marks the experimental temperature.

neighbours without compensating for this offset (Fig. S7). How does the hole-hole correlation map look like without post-selection (such as in Fig. 3)?

The referee is correct that the doping level has a strong impact on the observed signal, as one would expect. We study this effect in detail in Fig. 2c of the manuscript. As can be observed there, the interesting feature at distance $\mathbf{d} = (0, 2)$ is non-monotonous with doping and shows a maximum around $\delta = 0.18$. We attribute this behaviour to different competing effects: in the low-doping regime, our preparation does not produce enough dopants in every chain to allow for significant stripe formation, in addition to the suppression of charge ordering due to residual next-nearest neighbour hopping. At high doping, the dopants reduce the spin correlations within the system, which in turn lowers the effective attraction between dopants. Therefore, only in the intermediate doping regime we find significant stripe formation at larger distances. This then leads to the specific choice of doping level in Fig. 2a of the manuscript. For illustration, we show the hole correlations combining all doping levels in Fig. R2 of this reply, which mostly features the Pauli hole and nearest-neighbour hole-hole attraction along y with smaller positive signals at $d_y = 3 - 5$.

Figure R2: **Hole-hole correlations on full dataset.** The close-range attraction observed at $\mathbf{d} = (0, 1)$ remains clearly visible.

3. The correlation maps display additional features that are not discussed in the text, such as positive Y-shaped features in Fig. 2a above and below $\mathbf{d} = (0, 0)$, and positive pair correlators

at $|d_y| = 3, 4$ in Figs. 3b and c. Are these features a physical indication of fluctuations along x of the stripe position, or of edge effects? Or do they result from experimental limitations, such as potential inhomogeneities? These features are at least as statistically significant as the positive signal pointed out at $d_x = \pm 5$ and are therefore worth commenting on.

We agree with the referee that there are additional features that were not discussed in full detail in the manuscript as we were not fully convinced of their significance ourselves. We will describe them in more detail here.

The positive correlations at increasing d_y away from $d_x = 0$ are directly related to the fluctuations of charge structures along the perpendicular x direction. Due to the coupling along x , the individual stripes are not fully rigid but instead fluctuate due to hole movement along x . The Y -structure visible at large d_y is mostly a result of the single datapoint at $\mathbf{d} = (3, 6)$. Although statistically significant, it may be related to edge effects in the system such that we do not claim any additional pairing or stripe formation at these large distances.

Individually, the data-points at $d_x = 5$ are on the same significance level as the substructure at large $|d_y|$ but given that we see 3 adjacent positive correlations at $\mathbf{d} = (5, 0), (5, 1), (5, 2)$ enhances their combined significance (average d_y) to a level that convinced us to consider the possibility of having an additional neighbouring stripe.

The pair-pair and pair-hole correlations at $\mathbf{d} = (0, 3), (0, 4)$ are related to the extended charge structures that form within the system. The reason why correlations at these distances might exceed shorter distances may be attributed to the residual next-nearest-neighbour hopping present in the system.

In general, it is currently difficult to interpret the details of the correlation pattern without a high degree of uncertainty. However, it will be an exciting direction for a follow-up study.

4. On a related note, it would be important to provide quantitative estimates of the system homogeneity, and to state whether this affects the observed density correlations. For instance, a residual harmonic confinement or anti-confinement along x , together with total atom number fluctuations, could also explain long-range connected density correlations along the y direction. In addition, is the distribution of the stripes detected in Fig. 4 uniform along the x direction?

We show the density in Fig. R3a of this reply and find a residual density standard deviation of 0.039 within the system. These inhomogeneities are likely caused by the anticonfinement of our lattices, as well as a small imbalance between different chains due to the preparation procedure applying the potential offset (see also Fig. S4b of the manuscript). We do not apply any potential shaping with the DMD within this central system.

Nevertheless, as the referee points out, we use *connected* density-density correlators, which remove the effects of density inhomogeneities from the interpretation of our signal by construction. We furthermore argue that the preparation of the system is identical in the mixD and 2d cases. Hence, spurious correlations due to density inhomogeneities and atom number fluctuations, as the referee suggests, would show both in the mixD and 2d cases, which we do not observe (see Fig. 2b of the manuscript). We have added the density plot to the Supplementary Materials and note that no vertical structures are visible in this density.

As the referee suggested, we investigated the distribution of stripes along x in Fig. R3b of

this reply for a fixed length of $\ell = 4$ and at a doping of $\delta = 0.18$. We find small deviations which are similar to the observed density distribution.

Figure R3: **Local densities within the system.** **a**, Inhomogeneities of atomic densities are due to residual anticonfinement of our lattice beams as well as our preparation. **b**, Density of the center of mass of 'stripes' with length $\ell = 4$ at $\delta = 0.18$. The distribution reflects the overall density distribution of holes in **a**.

5. Additional comments are needed to clarify the claims of the existence of a second stripe or a charge density wave (Lines 164-167 and 228-232):

- (a) Fig. 2: Is the positive signal at \$d_x = \pm 5\$ visible at dopings other than \$\delta = 0.18\$? If it is, is its position consistent with the expected period of the charge density wave \$1/\delta\$?

Unfortunately, a full investigation of this signal across all doping ranges was not possible within this study. The main reason is the narrow parameter regime in which substantial extended charge correlations are present. However, at the discussed doping level of $\delta = 0.18$, we expect a periodicity $1/0.18 = 5.6$ sites, which is close to the measured position. Furthermore, to properly extract the period for a varying doping level, extended charge correlations at doping levels of $0.25 - 0.3$ would be required, where a reduction of the period towards 4 sites would be expected. At our temperature, this doping level is not favourable for us because of reduced spin correlations such that we did not observe a varying position of this second maximum with doping.

- (b) Fig. 2: Shouldn't the positive hole-hole correlations at \$d_x = \pm 5\$ entail the existence of negative correlations at \$d_y \neq 0\$ and \$1 \leq |d_x| \leq 4\$ due to Pauli repulsion between the two stripes?

We agree with the referee that, for fully formed static stripes, the intermediate region should exhibit negative correlations. Including fluctuations of the stripes along x , we expect even at low temperatures a strongly reduced contrast for $d_y \gtrsim 2$ when averaging over those fluctuations by using a 2-point correlator as correlations form a cone-like structure (see also the reply to comment #3). In our case, finite temperature also leads to lower probabilities for extended stripes to form. A distance $1/n^h$ between stripes with correlated fluctuations will only show up at small d_y when averaging over those fluctuations in this two-point correlator.

- (c) Figs. 3b and c: I do not really understand why the average of the correlators along \$y\$ is interpreted as the existence of a charge structure at \$d_x = 4\$. It seems here to result

from a fortuitous combination of strong Pauli repulsion on the central lines $|d_y| \leq 1$, which dominates the average at $|d_x| \leq 3$, and a broad positive background on the outer lines $|d_y| > 1$, which dominates the average at $|d_x| \geq 4$. Wouldn't a certain type of long-range anisotropic correlations without stripe order lead to a similar average? There is no proper density wave structure visible for any individual line, what about after binning with respect to doping like in Fig. 2?

We agree with the referee's conclusion that no definitive density wave structure can be extracted from our data. In fact, a combination of correlation values at different positions along y results in the observed positive signal. However, the main observation is a local maximum, regardless of its absolute value. This maximum is visible both at $|d_y| \leq 1$ (where the sign is still negative) and in the data averaging all d_y . Therefore, we only note that there are hints of a second stripe forming at this distance. A binning by doping is unfortunately not possible on our current dataset due to the smaller number of pairs that contribute to these correlators, which requires considerably higher statistics than what we were able to gather experimentally.

To summarise, we have adapted our manuscript following the referee's comments to put less emphasis on the possible presence of a second stripe.

6. Unless there is strong evidence for a second stripe, the plural word "stripes" in the title is potentially misleading, as it could be understood as the observation of several stripes in individual experimental realizations, instead of the observation of a single stripe in correlations measured over repeated experimental realizations.

Although we hope the updated abstract and introduction should help prevent misunderstanding, we understand the concerns of the referees. Accordingly, we are open to modifying the title of the manuscript by replacing the term "stripes" with "individual stripes".

7. As far as I understand, the spin-correlation function between next-nearest neighbours along x is compatible with what is expected for decoupled 1d chains with antiferromagnetic correlations (Ref. [51]). In contrast, negative spin correlations on diagonal neighbours around the hole are reminiscent of the spin texture of a magnetic polaron also observed in the same group (Ref. [38]). Are there indications of longer-range domains along the stripe direction in the hole-spin-spin correlations? For example, does the presence of a hole at $d = (0, 0)$ increase AFM correlations along vertical nearest-neighbour bonds $(1, j) \leftrightarrow (1, j+1)$?

We are grateful for the referee's insightful comment on the interplay of spin and charge degrees of freedoms. We agree with their statement concerning the relation to previous 1d and 2d measurements. Following their suggestion, we checked the nearest-neighbour spin bonds around the dopant (see Fig. R4 of this document). The connected correlator is slightly positive for $d^s = (0, 1)$ directly next to the hole. The reasons for this are found in the residual polaronic features, where fluctuations along x lead to a more positive correlation at this bond. However, in the mixD setting, such features are expected both from individual holes forming magnetic polarons as well as from holes bound into a stripe.

8. L.144, "Due to the finite size and particle number fluctuations": can the authors give more intuition on the systematic offset? In the absence of fluctuations, where exactly N atoms are independently and identically distributed over system size V , the conditional probability of

Figure R4: **Connected hole-spin-spin correlations for nearest-neighbouring bonds.** The bonds along y are positive, which is both a feature of individual holes forming magnetic polarons, and of extended fluctuating stripes.

finding a particle at any site j given a particle is located at site $i \neq j$ is reduced by $1/V$ (since $N-1$ particles, not N , are distributed over all sites other than i), which would lead to a negative g_2 -Function equal to $-1/N$? Conversely, one would expect that large particle number fluctuations to lead to positive long-range density-density correlations. These two limits could be more explicitly explained in Methods D2.

We appreciate the referee's interest in the offset correction. Their intuition is identical to our understanding of this effect. It is essentially a $-1/N$ correction in the case of a fixed number of randomly distributed hard-core particles. Similarly, a positive offset can occur if global fluctuations are so large that on average more than N particles remain after one is found at i . We added a sentence to the section in the Methods to clarify this point.

9. L.146: since the offset o_δ is doping-dependent, why not indicating a range o_δ -0.03 to -0.06 (according to Fig. S7b)? What is the experimental error on this offset?

We followed the referee's suggestion and now provide the offset's range. For the relevant data of Fig. 2, the doping is fixed, such that the offset is only given by the overall doping level and therefore precise. Even for larger doping ranges, e.g. $\delta = 0.17 - 0.2$, the offset is $o_\delta = -0.044$ with an uncertainty of 7×10^{-5} , such that the error is negligible.

10. L.280-282, "While our parameter regime is not favourable to investigate structure factors": could the authors clarify this statement?

We work in a parameter regime where $J_y \approx 4J_x$, clearly exceeding J_x . This leads to stronger spin correlations along y , which in turn leads to stronger effective attraction favouring charge correlations along y . However, the spin correlations along x are rather short-ranged (see also last comment 12), such that no large spin correlation domains are present in this direction for our temperatures. As domains on the order of the expected distance between stripes are a prerequisite to reveal features within the spin structure factor, we would require much lower temperatures for proper investigations using this observable. We have added a sentence in the SI to clarify our statement.

11. L.297-299 and L.325-329: is there any theory reference describing the "change in the parity of the AFM pattern across impurities"? Only Ref. [51] from the same group is cited.

We thank the referee for identifying this missing reference which we now added to the revised manuscript [5].

12. L.335-339, "we observe these features without long-range AFM correlations": could the authors indicate spin-spin correlation lengths along the x - and y -directions?

Our correlation lengths are comparably short for a number of reasons. First of all, in the doping regime around 20%, the interplay between the spin- and charge degrees reduces the absolute value of spin correlations significantly. Secondly, the strong interactions ($U/t_x = 27(2)$, $U/t_y = 17(1)$) lead to a low J less favourable for long-range spin correlations at our temperature. Finally, the strongly anisotropic couplings further reduce correlations along x as they bias correlations towards y . Therefore, we observe significant spin correlations only up to a distance of three sites (see Fig. R5). We added this information to the supplementary information of the manuscript.

Figure R5: **Spin correlations at further distances.** **a**, Spin correlation map at $\delta = 0.18$. Only close distances exhibit non-zero values. A cut along x (y) is shown in **b** in blue (red).

Response to report from the third referee

We appreciate the referee's positive feedback on our work. Their detailed comments on the numerical simulations are highly valuable in improving the quality of our manuscript.

1. It is not clear why the authors presented the DMRG for the largest lattices (8x4) in the SI, while in the main text they show results to smaller ones (8x3).

In this case, we are limited by the current capabilities of our finite-temperature DMRG calculations. We highlight that for finite-temperature calculations based on purification, the effective width of the simulated system doubles [6], which severely limits the accessible (physical) system width L_y particularly at finite doping. For the lowest doping of 12.5%, calculations for up to $L_y = 4$ were possible; higher doping levels, however, are only accessible for $L_y = 3$. To achieve convergence even for this modest system size, cutting-edge bond dimensions of $\chi_{\text{TDVP}} \sim 30000$ were required, highlighting the limits of state-of-the-art numerical methods in this context. Because most of our experimental data are at higher doping levels, we only compare to a variety of dopings at the smaller system size. We note that both system sizes can still be affected by finite-size effects.

2. When discussing Figure 2.d, the authors mention that the theoretical results agree "qualitatively" with the experimental ones. This claim is incompatible with Fig.2. For instance, the (0,2) case is negative for the experimental data, but positive for the DMRG ones. This is a clear qualitative and quantitative difference. The authors explain this inconsistency by mentioning that it may be due to finite-size effects. If so, I return to item 1: why was the largest lattice in the SI and not in the main text?

We thank the referee for helping us clarify the connection between the experimental data and the numerical simulations. While we agree that there are parameter regimes, where experimental and numerical results do not match, we note that finite-size effects are only one among multiples that could affect the differences between experiments and theory. Especially for the distance $d = (0, 2)$, the small next-nearest-neighbour coupling described in the SI (as well as being characterised in more detail in [3]), reduces the correlation at this particular distance in the experiment such that only in certain doping regions, the effective charge attraction dominates over the Pauli repulsion. A second significant point is the preparation. In the experiment, we do not prepare identical doping levels in every chain but instead have a statistical distribution of doping per chain with the mean doping stated in the text. If the number of holes in chain $y = i + 1$ is smaller than in $y = i$, there are necessarily holes without a partner. For those, we cannot expect a positive correlation signal in chain $i > 1$, so the mean correlator is reduced. This effect is not explicitly included in the theory. To avoid many cases with zero holes in some chains, we minimally require a doping level of around 15% used in the experiment. As mentioned, we chose the smaller $L_y = 3$ system for the DMRG calculations to match the doping.

Finally, the differences between numerical data and experimental results are mainly quantitative. This is very much expected, on the one hand, because of the limit to the numerical system size. On the other hand, our experiment simulates the Fermi-Hubbard model, while it is compared to numerics of the $t - J$ model. Even disregarding the next-nearest neighbour hopping term, there are additional processes that the $t - J$ model does not fully capture, leading to potential quantitative deviations.

- Suppose we agree that strong finite-size effects are leading to this quantitative disagreement on theoretical and experimental correlation functions (even for nearest neighbours), as shown in Fig.2, and as emphasized by the authors. Then, it leads me to another question: why should one believe that the temperature estimation by DMRG, provided in Fig.S2 (probably for an 8×3 lattice), has not suffered from strong finite-size effects? From the theoretical-experimental inconsistencies presented in Fig.2, my partial conclusion is that one has many reasons to believe that the temperature is not well defined in the paper, or at least its error bars should be much larger than those presented.

The referee is correct that finite-size effects may in principle affect our temperature estimation as well. However, the spin and charge sectors are not sensitive to the same degree on the size of the system. We compared DMRG spin correlations at $L_y = 3$ and $L_y = 4$ at $\delta = 0.125$ and find close agreement (see Fig. R6 below). Although this may not hold as well at higher doping values, previous studies on our experiment have shown that as a result of our preparation, the temperature does not change significantly for differing doping levels. Finally, we note that, for the reason the referee stated, we already provide relatively large error bars on our temperature estimate.

Figure R6: **DMRG spin correlations for different system sizes.** Comparing the nearest-neighbour DMRG spin-spin correlations along y for a system size of $L_y = 3$ at two different doping levels with a slightly larger system at $L_y = 4$.

- If the temperature estimation of DMRG is correct, it would be worth it if the authors could provide an estimation for the temperature at which the stripes appear in the 2D case. With this, the community would know clearly how challenging this task is.

We thank the referee for pointing out this valuable information that is so far not explicitly mentioned in the manuscript. Although we again note that theoretical calculations may be subject to finite-size limitations, simulations using maximally entangled typical thermal states (METTS) have shown that stripes in 2d systems appear at around $0.05 t$ [4] which we now state more explicitly in the text. For larger systems, slightly lower temperatures might be required to observe stripe order. However, these results mostly focused on spin and charge structure factors such that correlations might already be visible at slightly higher temperatures. Finally, these results were achieved with isotropic spin couplings, which might not be ideal to realize highest possible transition temperatures.

5. The authors mention in the text that the engineering of the mixD setting just "slightly" increases the spin coupling in the perpendicular direction. However, from the data reported in the SI the one has $J_y/J_x \approx 4$. It may be a slight increase in J_y , but the final exchange anisotropy is large. I recommend removing the "slightly" in the phrase, to avoid misleading conclusions.

We adapted the manuscript according to the referee's suggestion.

6. The acronym DRMG is not defined in the main text.

We added the definition to the main text.

References

- [1] Sun, H. *et al.* Signatures of superconductivity near 80 K in a nickelate under high pressure. *Nature* **621**, 493–498 (2023).
- [2] Schlömer, H., Bohrdt, A., Pollet, L., Schollwöck, U. & Grusdt, F. Robust stripes in the mixed-dimensional $t - J$ model. *Phys. Rev. Res.* **5**, L022027 (2023).
- [3] Chalopin, T. *et al.* Optical superlattice for engineering Hubbard couplings in quantum simulation. In preparation (2024).
- [4] Wietek, A., He, Y.-Y., White, S. R., Georges, A. & Stoudenmire, E. M. Stripes, Antiferromagnetism, and the Pseudogap in the Doped Hubbard Model at Finite Temperature. *Phys. Rev. X* **11**, 031007 (2021).
- [5] Zaanen, J. CURRENT IDEAS ON THE ORIGIN OF STRIPES. *J. Phys. Chem. Solids* **59**, 1769 (1989).
- [6] Nocera, A. & Alvarez, G. Symmetry-conserving purification of quantum states within the density matrix renormalization group. *Phys. Rev. B* **93**, 045137 (2016).

Reviewer Reports on the First Revision:

Referee #1 (Remarks to the Author):

I'm satisfied by the response of the author. The experimental study of a mixed-dimensional Hubbard model is quite amazing. I recommend the paper to be published in Nature.

Referee #2 (Remarks to the Author):

I would like to thank the authors for their detailed answer and changes to the manuscript, which addressed my questions. I appreciated the way the anisotropic mixD system and nickelates are now explicitly connected in the first part, as well as the extra discussion of temperature scales and experimental data. I would recommend publication in Nature, provided that the few following points prompted by the authors' reply are properly addressed:

1. I recommend using the phrase "individual stripes" in the title, as suggested in point 6, since the authors agree that the evidence for multiple stripes is weak.
2. I hadn't realized that the system of interest is actually restricted to a size of about 50 sites according to Fig. R3, consistently with the half-size 7x7 of the hole-hole correlation maps. Although this 2d system is still challenging to simulate numerically, its experimental size is comparable to the four 2x7 decoupled ladders realized in the authors' previous work [46], which itself came with its own additional challenges (e.g. potential shaping). It would be great if the authors could highlight in the manuscript the key experimental advances that enabled this work, beyond simply going to a two-dimensional system. The typical size of the region of interest should also be clearly stated in the text.
3. As pointed out by Referee #1, long stripes are exceedingly rare and in the doping and temperature ranges explored in this work, length-3 and 4 stripes are the significant ones that distinguish experimental histograms from a random distribution (Fig. 4). In this respect, the length-10 stripe in Fig. 1c seems cherry-picked and a more statistically representative snapshot such as the inset of Fig. 4a should be used instead to more honestly show the type of long-range hole structures realized in this work.
4. The new Fig. S1b shows some staggering of the density along y (hard to discern right now because of the colorbar extents), presumably due to finite imbalance Δ . It would be great to state the typical density imbalance and briefly comment on the validity of the uniform Hamiltonian (1).

Referee #3 (Remarks to the Author):

I thank the authors for their response to my queries.

As I mentioned in my first report, the experimental results are solid and sound, and despite the concerns raised by the other referees, I believe they make an important step toward a deeper understanding of stripe formation. However, I remain unconvinced about the numerical data. The authors claim that the numerical results are only "qualitatively" different from the experimental ones, and I reinforce that it is misleading, as illustrated in Fig. 2. I cannot agree that correlation functions with opposite signs are "qualitatively" different. It seems that, due to the strong finite-size effects, the authors are not able to present more accurate data. I also do not agree with their argument that the possible difference could be due to the use of the t-J model instead of the Fermi-Hubbard model. If so, why did they simulate the t-J model rather than the Fermi-Hubbard one?

I agree that cold atom experiments provide an excellent platform for exploring the fundamental properties of matter. However, I have concerns about the accuracy of the numerical data presented in this manuscript. Inaccurate (or misleading) results should not be published in high-impact journals such as Nature. Therefore, I do not recommend this paper for publication in Nature, but I suggest considering one of its sister journals instead.

Referee #4 (Remarks to the Author):

The manuscript by Bourgund et al. presents results of an experimental study of the Fermi-Hubbard model in two dimensions in the strong-coupling region. In their realization of the model, they impose a superlattice potential along the y direction that drives the tunneling in that direction to near zero while preserving the exchange interaction. Thinking in terms of an equivalent (anisotropic) t - J model, the exchange interaction along y would also be about four times larger than the exchange interaction along x . The goal is to promote bunching of holes along the y direction (in appropriately doped systems) since Pauli blocking is largely suppressed in that direction. In fact, they observe this phenomenon through various two- and three-point correlation functions. Those point to stripe-like patterns of 3-5 sites in length that appear with a likelihood roughly twice as much as that for an isotropic 2D system or a system with randomly placed holes near $1/8$ doping. Even though the system is engineered to avoid Pauli blocking in one direction (and therefore, favor such patterns), the results are far from trivial and quite interesting in light of recent numerical studies suggesting stripes as the ground state of the isotropic model. As far as I know, this work represents the first study of stripe physics in a 2D Hubbard model with cold fermionic atoms.

I have also read reports of the previous three reviewers and the authors' response to them. Firstly, I understand the third referee's frustration (for lack of a better word) with the theoretical results. At first glance, they do not appear to be the state of the art. For example in Ref. 17, DMRG studies of Hubbard systems as large as 48×6 with bond dimensions larger than 20,000 were carried out. However, the latter were at zero temperature and I suspect that time evolution to achieve finite temperatures in this work significantly adds to the computational toll. This can perhaps be more clearly conveyed for an average reader in the SI using non-technical language.

Secondly, I agree with the third referee that the departure of numerical results from the experimental data in Fig. 2 regarding the hole-hole correlation can be characterized as more than quantitative. Let me be more specific: I do consider the trend in the correlations at $d=(0,1)$ and $(1,1)$ (orange and purple data in Fig. 2d) qualitatively the same as those observed in the data (Fig. 2c); both show an overall decrease with increasing the doping. The correlation at $d=(0,2)$, on the other hand, exhibits the opposite trend and is not qualitatively comparable with the experimental data, as stated in the revised manuscript, in my opinion. However, to me, this is a matter of further revising the language in the text. If the authors are willing to edit the last paragraph of Sec. IV (see my suggestion below*), my recommendation would be to publish in Nature given the level of interest in the topic among the community and the quality and thoroughness of the work.

*While the trend in correlations at $d=(0,1)$ and $(1,1)$ is qualitatively comparable to the experimental data, the same cannot be stated about correlations at $d=(0,2)$. We attribute these differences to ... (also add simulating the t - J model with DMRG while the experimentally realized system is Hubbard in nature).

Other comments and suggestions (in the order they appear in the manuscript):

"mixD" is not defined when first mentioned.

"This leads to an increase in the characteristic energy scales of collective effects in<-- experimentally accessible regimes"(?)

"In nickelates, which are mixD-bilayer systems, this causes high critical temperatures for superconductivity"

Is this established as a causation or could it be a correlation?

In line 120, I am not clear what is meant by 'key concepts' or the apparent distinction between charge ordering and stripes in this context. Perhaps consider reformulating the sentence.

I believe the estimate for J_x/t_x (~ 0.15) should also be provided after Eq. (1), especially since estimates for every other Hamiltonian parameter (t_x, t_y, J_y) is provided. I find the t_y^{2d} notation a bit confusing since it looks as if one is raising t_y to the power of $2d$. Why not simply use t'_y for example?

N_d is called the normalization after Eq. (2), but a definition is not given. I assume it is the number of bonds with length d .

In the last sentence of the third paragraph of Sec. IV (line 188), I am not sure what the statement is referring to; does it refer to the difference between $2d$ and MixD or between x and y directions in $2d$? About the latter, what I actually see in the inset of Fig. 2b is that the antibunching along x is reduced (less negative) compared to those along y . And, why should the $g(2)$ function be anisotropic for the standard $2d$ system in the first place?

In the next paragraph (line 196), it is stated that both nearest-neighbour and diagonal correlations decrease with doping. However technically, the nearest-neighbour correlations increase first before showing a downward trend as doping increases. So, the statement should be modified accordingly in my opinion.

In the last paragraph of Sec. IV, I would mention the value of J_x/t_x too. It is currently given in the SI.

In Fig. 3, top panels in b and c, maybe add a bar on top of $g(2)$ functions to indicate averages along y . I was initially confused just looking at the figure, thinking the top panel in c for example shows the quantity along the dashed line cut in the map below it.

"This reveals a slightly positive signal at a distance of $dx = 4$, similar to Fig. 2."

This is somewhat of an overstatement to me because Fig. 2 shows positive correlations at $dx=5$ for $dy=0$, whereas in this case, the positivity comes from $dy \neq 0$.

Is there a reason why uncorrelated parts are not subtracted from the quantity in Eq. (5)? I am not sure how to interpret the trends in Fig. 5 for that reason.

I would have liked to see more information about the theory results in the caption of Fig. 5. Are they obtained using 8x3 or 8x4 systems? Why is it that more dopings could be represented here compared to Fig. 2?

In line 347 (page 5), it is unclear to what "both" refers. It would be clearer if the regular spin-spin correlation was mentioned by name.

TDVP is not defined in the SI.

Author Rebuttals to First Revision:

Introductory statement

We thank the referees for their valuable feedback on our work. We followed their suggestions on a change of the title as well as a clarification on the theory comparison, and improved the manuscript with the changes listed below.

Changes to the Manuscript

- Changed title to Formation of individual stripes in a mixed-dimensional cold-atom Fermi-Hubbard system
- Added definition of mixed-dimensional
- Changed exemplary shot in Fig. 1c following referee 2's suggestion
- Added precise parameters of J_x/t_x to the main text
- Changed notation of t_y^{2d}
- Improved phrasing on theory comparison of Fig. 2
- Added explanation on normalisation in Eq. 2
- Changed notation of y-averaged correlator in Fig 3b/c
- Added system size used in numerical calculations to caption of Fig. 5

Changes to the Methods

- Added comment on residual imbalance
- Added definition of TDVP
- Added comment on difference in system size of numerical simulations at zero and finite temperature
- Updated arXiv reference
- Combined figures to reduce number of extended data figures

Response to report from the second referee

We thank the referee for their appreciation of our work. We addressed their remaining points as listed in the following:

1. I recommend using the phrase "individual stripes" in the title, as suggested in point 6, since the authors agree that the evidence for multiple stripes is weak.

We followed the referee's suggestion and adapted our title accordingly.

2. I hadn't realized that the system of interest is actually restricted to a size of about 50 sites according to Fig. R3, consistently with the half-size 7×7 of the hole-hole correlation maps. Although this 2d system is still challenging to simulate numerically, its experimental size is comparable to the four 2×7 decoupled ladders realized in the authors' previous work [46], which itself came with its own additional challenges (e.g. potential shaping). It would be great if the authors could highlight in the manuscript the key experimental advances that enabled this work, beyond simply going to a two-dimensional system. The typical size of the region of interest should also be clearly stated in the text.

The referee refers to a system size of about 50 sites, which is however only the central, rectangular part of our system. In total, we use 109 sites in a circular region of interest for the calculation of our correlators. We state the number of sites in the main text and show the full experimental density in the SI.

Comparing this experiment to our previous work on 2×7 ladders, we now use a very different approach to achieve the required potential offset between neighbouring chains. Instead of relying on local potential shaping, we make use of the tunability of our optical superlattices to generate staggered potentials. We refer to the sections on the Experimental sequence as well as Offset phase calibration for further details. Finally, more technical information is given in our more recent work [1].

3. As pointed out by Referee 1, long stripes are exceedingly rare and in the doping and temperature ranges explored in this work, length-3 and 4 stripes are the significant ones that distinguish experimental histograms from a random distribution (Fig. 4). In this respect, the length-10 stripe in Fig. 1c seems cherry-picked and a more statistically representative snapshot such as the inset of Fig. 4a should be used instead to more honestly show the type of long-range hole structures realized in this work.

We followed the referee's suggestion and exchanged the exemplary shot in Fig 1c to a more common realisation.

4. The new Fig. S1b shows some staggering of the density along y (hard to discern right now because of the colorbar extents), presumably due to finite imbalance Δ . It would be great to state the typical density imbalance and briefly comment on the validity of the uniform Hamiltonian (1).

The referee is correct that there is some residual imbalance within our system. We now state the mean normalised imbalance as 0.037 in the text (consistent with the value found in Fig. S4b at the main experimental parameters). However, as a result from this imbalance the validity of the t - J Hamiltonian is unaffected as the couplings remain the same. Instead, we only expect a contribution to our finite doping resolution.

Response to report from the third referee

We thank the referee for their positive feedback on the experimental results. We understand the referee's concerns on the differences between experimental and theoretical results. As nicely pointed out by referee four, there are correlation distances where theory and experiment do indeed qualitatively agree. Meanwhile the distance $d = (0, 2)$ does show significant differences. As mentioned in the text, this is partially caused by the residual next-nearest neighbour coupling not captured by the t-J model. Therefore we now follow the guidance by referee four and improved the interpretation of the comparison. We would like to emphasize the limits of the numerical calculations in system size, despite being state of the art. Simulating the Fermi-Hubbard model instead of the t-J model, as the referee suggests, was unfortunately not possible as this leads to even smaller numerical system sizes.

Response to report from the fourth referee

We thank the referee for their insightful comments and helpful suggestions. We followed their advice which greatly helped in improving the manuscript. We address their individual comments in the following:

I have also read reports of the previous three reviewers and the authors' response to them. Firstly, I understand the third referee's frustration (for lack of a better word) with the theoretical results. At first glance, they do not appear to be the state of the art. For example in Ref. 17, DMRG studies of Hubbard systems as large as 48×6 with bond dimensions larger than 20,000 were carried out. However, the latter were at zero temperature and I suspect that time evolution to achieve finite temperatures in this work significantly adds to the computational toll. This can perhaps be more clearly conveyed for an average reader in the SI using non-technical language.

We fully agree with the referee. Performing finite temperature calculations is significantly more challenging than zero-temperature results, leading to limits in the achievable system size. Following the referee's suggestions, we clarified this issue in the SI.

Secondly, I agree with the third referee that the departure of numerical results from the experimental data in Fig. 2 regarding the hole-hole correlation can be characterized as more than quantitative. Let me be more specific: I do consider the trend in the correlations at $d=(0,1)$ and $(1,1)$ (orange and purple data in Fig. 2d) qualitatively the same as those observed in the data (Fig. 2c); both show an overall decrease with increasing the doping. The correlation at $d=(0,2)$, on the other hand, exhibits the opposite trend and is not qualitatively comparable with the experimental data, as stated in the revised manuscript, in my opinion. However, to me, this is a matter of further revising the language in the text. If the authors are willing to edit the last paragraph of Sec. IV (see my suggestion below*), my recommendation would be to publish in Nature given the level of interest in the topic among the community and the quality and thoroughness of the work.

*While the trend in correlations at $d=(0,1)$ and $(1,1)$ is qualitatively comparable to the experimental data, the same cannot be stated about correlations at $d=(0,2)$. We attribute these differences to ... (also add simulating the t-J model with DMRG while the experimentally realized system is Hubbard in nature).

We thank the referee for their insights. We agree that the qualitative similarity is restricted to $d = (0, 1), (1, 1)$ while $d = (0, 2)$ behaves significantly differently due to number of reasons discussed in the text. We appreciate the referee's suggestion for an improved phrasing of this paragraph and followed their advice.

Other comments and suggestions (in the order they appear in the manuscript):

1. "mixD" is not defined when first mentioned.

We added the definition in the beginning of the main text.

2. "This leads to an increase in the characteristic energy scales of collective effects \rightarrow in \leftarrow experimentally accessible regimes" (?)

By using mixD systems, we change the energy scales such that collective effects are now experimentally accessible. We adjusted the sentence to improve the clarity: "This leads to an

increase in the characteristic energy scales of collective effects as kinetic and magnetic terms in the Hamiltonian are less frustrated elevating these effects to experimentally accessible regimes."

3. "In nickelates, which are mixD-bilayer systems, this causes high critical temperatures for superconductivity" Is this established as a causation or could it be a correlation?

The experimental studies of superconductivity in bilayer nickelates under pressure, presumed to be described by an effective model with mixed dimensionality, is still in its infancy. As such, the observation of remarkable high critical T_c 's in these systems should best be viewed as an experimentally established correlation. There are several theoretical works in the literature arguing for a causal connection between mixD and high T_c 's, however further experimental results will be needed to corroborate those ideas in the context of these specific materials.

4. In line 120, I am not clear what is meant by 'key concepts' or the apparent distinction between charge ordering and stripes in this context. Perhaps consider reformulating the sentence.

We thank the referee for pointing out this inaccuracy as the concept we referred to was charge ordering i.e. individual stripes. We changed 'key concepts' to 'key concept' and hope this clarifies the issue.

5. I believe the estimate for J_x/t_x (~ 0.15) should also be provided after Eq. (1), especially since estimates for every other Hamiltonian parameter (t_x, t_y, J_y) is provided. I find the t_y^{2d} notation a bit confusing since it looks as if one is raising t_y to the power of $2d$. Why not simply use t'_y for example?

We added the estimate of J_x/t_x to the main text and changed the notation for t_y^{2d} to t'_y .

6. N_d is called the normalization after Eq. (2), but a definition is not given. I assume it is the number of bonds with length d .

The referee is correct and we clarified it in the main text.

7. In the last sentence of the third paragraph of Sec. IV (line 188), I am not sure what the statement is referring to; does it refer to the difference between 2d and MixD or between x and y directions in 2d? About the latter, what I actually see in the inset of Fig. 2b is that the antibunching along x is reduced (less negative) compared to those along y . And, why should the $g(2)$ function be anisotropic for the standard 2d system in the first place?

We thank the referee for pointing towards this missing explanation. In this case, we refer to the change in x and y correlations when going from 2d to mixD. Initially, we find anticorrelations along both directions in 2d. They are not perfectly isotropic due to the anisotropic couplings which favour the y direction. When removing t_y by going into mixD, we thereby find more negative correlations along x instead. We improved the phrasing of this sentence for better clarity.

8. In the next paragraph (line 196), it is stated that both nearest-neighbour and diagonal correlations decrease with doping. However technically, the nearest-neighbour correlations increase first before showing a downward trend as doping increases. So, the statement should be modified accordingly in my opinion.

We agree with the referee that a clear decrease is only visible starting at $\delta \approx 0.15$, while below the nearest neighbour signal the error bars are large and the signal rather stays constant or increases with doping. Therefore, we clarified this statement in the text.

9. In the last paragraph of Sec. IV, I would mention the value of J_x/t_x too. It is currently given in the SI.

The value has been added to the main text.

10. In Fig. 3, top panels in b and c, maybe add a bar on top of $g(2)$ functions to indicate averages along y . I was initially confused just looking at the figure, thinking the top panel in c for example shows the quantity along the dashed line cut in the map below it.

We followed the referee's suggestion and changed the label accordingly.

11. "This reveals a slightly positive signal at a distance of $dx = 4$, similar to Fig. 2." This is somewhat of an overstatement to me because Fig. 2 shows positive correlations at $dx=5$ for $dy=0$, whereas in this case, the positivity comes from $dy \neq 0$.

We improved the phrasing to focus more on the qualitative similarities in the signals between Fig. 2 and 3 while not claiming this to be proof of a charge density wave.

12. Is there a reason why uncorrelated parts are not subtracted from the quantity in Eq. (5)? I am not sure how to interpret the trends in Fig. 5 for that reason.

We thank the referee for pointing towards this detail in our analysis. The full and connected correlators are related to slightly different physical interpretations, both of which are interesting. Here we opted to show the full correlator in the main text, while the connected correlator can be found in the SI (Fig. S6). By using the full correlator, we reveal that the effect of a dopant is strong enough to overcome the antiferromagnetic spin background and flip the sign of the correlator, as it is expected for spin stripe order. Therefore it allows us to compare the strength of the connected correlator to the strength of the antiferromagnetic background. Meanwhile, using the connected correlator might be more useful to disentangle the different contributions and investigate them individually, without comparing them to each other immediately.

13. I would have liked to see more information about the theory results in the caption of Fig. 5. Are they obtained using 8×3 or 8×4 systems? Why is it that more dopings could be represented here compared to Fig. 2?

The referee is exactly correct that the doping levels available differ between Fig. 2 and 5, while both are obtained on 8×3 systems (which we now clarify in the caption). The reason for this is related to the fact that different observables are investigated. Specifically, we find that pure charge correlations, such as the hole-hole correlator of Fig. 2 is numerically more challenging compared to the string correlator shown in Fig. 5b. This allowed us to include more doping levels in Fig. 5b.

14. In line 347 (page 5), it is unclear to what "both" refers. It would be clearer if the regular spin-spin correlation was mentioned by name.

We clarified the sentence to more explicitly refer to both correlators.

15. TDVP is not defined in the SI.

We added the definition to the SI.

References

- [1] Chalopin, T. Bojović, P. Bourgund, D. Wang, S. Franz, T. Bloch, I. Hilker, T. A. Optical superlattice for engineering Hubbard couplings in quantum simulation. arXiv:2405.19322 (2024).